# Estimating counterfactual treatment outcomes over time through adversarially balanced representations

**Ioana Bica**
Department of Engineering Science
University of Oxford, Oxford, UK
The Alan Turing Institute, London, UK
`ioana.bica@eng.ox.ac.uk`

**Ahmed M. Alaa**
Department of Electrical Engineering
University of California, Los Angeles, USA
`ahmedmalaa@ucla.edu`

**James Jordon**
Department of Engineering Science
University of Oxford, Oxford, UK
`james.jordon@wolfson.ox.ac.uk`

**Mihaela van der Schaar**
University of Cambridge, Cambridge, UK
University of California, Los Angeles, USA
The Alan Turing Institute, London, UK
`mv472@cam.ac.uk`

## Abstract

Identifying when to give treatments to patients and how to select among multiple treatments over time are important medical problems with a few existing solutions. In this paper, we introduce the Counterfactual Recurrent Network (CRN), a novel sequence-to-sequence model that leverages the increasingly available patient observational data to estimate treatment effects over time and answer such medical questions. To handle the bias from time-varying confounders, covariates affecting the treatment assignment policy in the observational data, CRN uses domain adversarial training to build balancing representations of the patient history. At each timestep, CRN constructs a treatment invariant representation which removes the association between patient history and treatment assignments and thus can be reliably used for making counterfactual predictions. On a simulated model of tumour growth, with varying degree of time-dependent confounding, we show how our model achieves lower error in estimating counterfactuals and in choosing the correct treatment and timing of treatment than current state-of-the-art methods.

## 1 Introduction

As clinical decision-makers are often faced with the problem of choosing between treatment alternatives for patients, reliably estimating their effects is paramount. While clinical trials represent the gold standard for causal inference, they are expensive, have a few patients and narrow inclusion criteria (Booth & Tannock, 2014). Leveraging the increasingly available observational data about patients, such as electronic health records, represents a more viable alternative for estimating treatment effects.

A large number of methods have been proposed for performing causal inference using observational data in the static setting (Johansson et al., 2016; Shalit et al., 2017; Alaa & van der Schaar, 2017; Li & Fu, 2017; Yoon et al., 2018; Alaa & van der Schaar, 2018; Yao et al., 2018) and only a few methods address the longitudinal setting (Xu et al., 2016; Roy et al., 2016; Soleimani et al., 2017; Schulam & Saria, 2017; Lim et al., 2018). However, estimating the effects of treatments over time poses unique opportunities such as understanding how diseases evolve under different treatment plans, how individual patients respond to medication over time, but also which are optimal timings for assigning treatments, thus providing new tools to improve clinical decision support systems.

The biggest challenge when estimating the effects of time-dependent treatments from observational data involves correctly handling the time-dependent confounders: patient covariates that are affected by past treatments which then influence future treatments and outcomes (Platt et al., 2009). For instance, consider that treatment A is given when a certain patient covariate (e.g. white blood cell

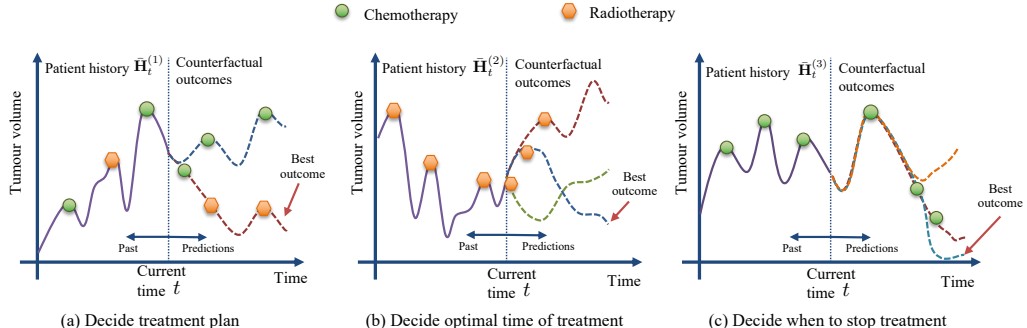

Figure 1: Applicability of CRN in cancer treatment planning. We illustrate 3 patients with different covariate and treatment histories $\bar{\mathbf{H}}_t$. For a current time $t$, CRN can predict counterfactual trajectories (the coloured dashed branches) for planned treatments in the future. Through the counterfactual predictions, we can decide which treatment plan results in the best patient outcome (in this case, the lowest tumour volume). This way, CRN can be used to perform all of the following: choose optimal treatments (a), find timing when treatment is most effective (b) decide when to stop treatment (c).

count) has been outside of normal range values for several consecutive timesteps. Suppose also that this patient covariate was itself affected by the past administration of treatment B. If these patients are more likely to die, without adjusting for the time-dependent confounding (e.g. the changes in the white blood cell count over time), we will incorrectly conclude that treatment A is harmful to patients. Moreover, estimating the effect of a different sequence of treatments on the patient outcome would require not only adjusting for the bias at the current step (in treatment A), but also for the bias introduced by the previous application of treatment B.

Existing methods for causal inference in the static setting cannot be applied in this longitudinal setting since they are designed to handle the cross-sectional set-up, where the treatment and outcome depend only on a static value of the patient covariates. If we consider again the above example, these methods would not be able to model how the changes in patient covariates over time affect the assignment of treatments and they would also not be able to estimate the effect of a sequence of treatments on the patient outcome (e.g. sequential application of treatment A followed by treatment B). Different models that can handle these temporal dependencies in the observational data and varying-length patient histories are needed for estimating treatment effects over time.

Time-dependent confounders are present in observational data because doctors follow policies: the history of the patients' covariates and the patients' response to past treatments are used to decide future treatments (Mansournia et al., 2012). The direct use of supervised learning methods will be biased by the treatment policies present in the observational data and will not be able to correctly estimate counterfactuals for different treatment assignment policies.

Standard methods for adjusting for time-varying confounding and estimating the effects of time-varying exposures are based on ideas from epidemiology. The most widely used among these are Marginal Structural Models (MSMs) (Robins et al., 2000; Mansournia et al., 2012) which use the inverse probability of treatment weighting (IPTW) to adjust for the time-dependent confounding bias. Through IPTW, MSMs create a pseudo-population where the probability of treatment does not depend on the time-varying confounders. However, MSMs are not robust to model misspecification in computing the IPTWs. MSMs can also give high-variance estimates due to extreme weights; computing the IPTW involves dividing by probability of assigning a treatment conditional on patient history which can be numerically unstable if the probability is small.

We introduce the Counterfactual Recurrent Network (CRN), a novel sequence-to-sequence architecture for estimating treatment effects over time. CRN leverages recent advances in representation learning (Bengio et al., 2012) and domain adversarial training (Ganin et al., 2016) to overcome the problems of existing methods for causal inference over time. Our main contributions are as follows.

**Treatment invariant representations over time.** CRN constructs treatment invariant representations at each timestep in order to break the association between patient history and treatment assignment and thus removes the bias from time-dependent confounders. For this, CRN uses domain adversarial training (Ganin et al., 2016; Li et al., 2018; Sebag et al., 2019) to trade-off between build-

ing this balancing representation and predicting patient outcomes. We show that these representations remove the bias from time-varying confounders and can be reliably used for estimating counterfactual outcomes. This represents the first work that introduces ideas from domain adaptation to the area of estimating treatment effects over time. In addition, by building balancing representations, we propose a novel way of removing the bias introduced by time-varying confounders.

**Counterfactual estimation of future outcomes.** To estimate counterfactual outcomes for treatment plans (and not just single treatments), we integrate the domain adversarial training procedure as part of a sequence-to-sequence architecture. CRN consists of an encoder network which builds treatment invariant representations of the patient history that are used to initialize the decoder. The decoder network estimates outcomes under an intended sequence of future treatments, while also updating the balanced representation. By performing counterfactual estimation of future treatment outcomes, CRN can be used to answer critical medical questions such as deciding when to give treatments to patients, when to start and stop treatment regimes, and also how to select from multiple treatments over time. We illustrate in Figure 1 the applicability of our method in choosing optimal cancer treatments.

In our experiments, we evaluate CRN in a realistic set-up using a model of tumour growth (Geng et al., 2017). We show that CRN achieves better performance in predicting counterfactual outcomes, but also in choosing the right treatment and timing of treatment than current state-of-the-art methods.

## 2 RELATED WORK

We focus on methods for estimating treatment effects over time and for building balancing representations for causal inference. A more in-depth review of related work is in Appendix A.

**Treatment effects over time.** Standard methods for estimating the effects of time-varying exposures were first developed in the epidemiology literature and include the g-computation formula, Structural Nested Models and Marginal Structural Models (MSMs) (Robins, 1986; 1994; Robins et al., 2000; Robins & Hernán, 2008). Originally, these methods have used predictors performing logistic/linear regression which makes them unsuitable for handling complex time-dependencies (Hernán et al., 2001; Mansournia et al., 2012; Mortimer et al., 2005). To address these limitations, methods that use Bayesian non-parametrics or recurrent neural networks as part of these frameworks have been proposed. (Xu et al., 2016; Roy et al., 2016; Lim et al., 2018).

To begin with, Xu et al. (2016) use Gaussian processes to model discrete patient outcomes as a generalized mixed-effects model and uses the $g$-computation method to handle time-varying confounders. Soleimani et al. (2017) extend the approach in Xu et al. (2016) to the continuous time-setting and model treatment responses using linear time-invariant dynamical systems. Roy et al. (2016) use Dirichlet and Gaussian processes to model the observational data and estimate the IPTW in Marginal Structural Models. Schulam & Saria (2017) build upon work from Lok et al. (2008); Arjas & Parner (2004) and use marked point processes and Gaussian processes to learn causal effects in continuous-time data. These Bayesian non-parametric methods make strong assumptions about model structure and consequently cannot handle well heterogeneous treatment effects arising from baseline variables (Soleimani et al., 2017; Schulam & Saria, 2017) and multiple treatment outcomes (Xu et al., 2016; Schulam & Saria, 2017).

The work most related to ours is the one of Lim et al. (2018) which improves on the standard MSMs by using recurrent neural networks to estimate the inverse probability of treatment weights (IPTWs). Lim et al. (2018) introduces Recurrent Marginal Structural Networks (RMSNs) which also use a sequence-to-sequence deep learning architecture to forecast treatment responses in a similar fashion to our model. However, RMSNs require training additional RNNs to estimate the propensity weights and does not overcome the fundamental problems with IPTWs, such as the high-variance of the weights. Conversely, CRN takes advantage of the recent advances in machine learning, in particular, representation learning to propose a novel way of handling time-varying confounders.

**Balancing representations for treatment effect estimation.** Balancing the distribution of control and treated groups has been used for counterfactual estimation in the static setting. The methods proposed in the static setting for balancing representations are based on using discrepancy measures in the representation space between treated and untreated patients, which do not generalize to multiple treatments (Johansson et al., 2016; Shalit et al., 2017; Li & Fu, 2017; Yao et al., 2018). Moreover, due to the sequential assignment of treatments in the longitudinal setting, and due to the change of

patient covariates over time according to previous treatments, the methods for the static setting are not directly applicable to the time-varying setting (Hernán et al., 2000; Mansournia et al., 2012).

## 3 PROBLEM FORMULATION

Consider an observational dataset $\mathcal{D} = \left\{ \{\mathbf{x}_t^{(i)}, \mathbf{a}_t^{(i)}, \mathbf{y}_{t+1}^{(i)}\}_{t=1}^{T^{(i)}} \cup \{\mathbf{v}^{(i)}\} \right\}_{i=1}^{N}$ consisting of information about $N$ independent patients. For each patient $(i)$, we observe time-dependent covariates $\mathbf{X}_t^{(i)} \in \mathcal{X}_t$, treatment received $\mathbf{A}_t^{(i)} \in \{A_1, \ldots A_K\} = \mathcal{A}$ and outcomes $\mathbf{Y}_{t+1}^{(i)} \in \mathcal{Y}_{t+1}$ for $T^{(i)}$ discrete timesteps. The patient can also have baseline covariates $\mathbf{V}^{(i)} \in \mathcal{V}$ such as gender and genetic information. Note that the outcome $\mathbf{Y}_{t+1}^{(i)}$ will be part of the observed covariates $\mathbf{X}_{t+1}^{(i)}$. For simplicity, the patient superscript $(i)$ will be omitted unless explicitly needed.

We adopt the potential outcomes framework proposed by (Neyman, 1923; Rubin, 1978) and extended by (Robins & Hernán, 2008) to account for time-varying treatments. Let $\mathbf{Y}[\bar{\mathbf{a}}]$ be the potential outcomes, either factual or counterfactual, for each possible course of treatment $\bar{\mathbf{a}}$. Let $\bar{\mathbf{H}}_t = (\bar{\mathbf{X}}_t, \bar{\mathbf{A}}_{t-1}, \mathbf{V})$ represent the history of the patient covariates $\bar{\mathbf{X}}_t = (\mathbf{X}_1, \ldots, \mathbf{X}_t)$, treatment assignments $\bar{\mathbf{A}}_t = (\mathbf{A}_1, \ldots, \mathbf{A}_t)$ and static features $\mathbf{V}$. We want to estimate:

$$\mathbb{E}(\mathbf{Y}_{t+\tau}[\bar{\mathbf{a}}(t, t+\tau-1)]|\bar{\mathbf{H}}_t), \tag{1}$$

where $\bar{\mathbf{a}}(t, t+\tau-1) = [\mathbf{a}_t, \ldots \mathbf{a}_{t+\tau-1}]$ represents a possible sequence of treatments from timestep $t$ just until before the potential outcome $\mathbf{Y}_{t+\tau}$ is observed. We make the standard assumptions (Robins et al., 2000; Lim et al., 2018) needed to identify the treatment effects: consistency, positivity and no hidden confounders (sequential strong ignorability). See Appendix B for more more details.

## 4 COUNTERFACTUAL RECURRENT NETWORK

The observational data can be used to train a supervised learning model to forecast: $\mathbb{E}(\mathbf{Y}_{t+\tau} \mid \bar{\mathbf{A}}(t, t+\tau-1) = \bar{\mathbf{a}}(t, t+\tau-1), \bar{\mathbf{H}}_t)$. However, without adjusting for the bias introduced by time-varying confounders, this model cannot be reliably used for making causal predictions (Robins et al., 2000; Robins & Hernán, 2008; Schulam & Saria, 2017). The Counterfactual Recurrent Network (CRN) removes this bias through domain adversarial training and estimates the counterfactual outcomes $\mathbb{E}(\mathbf{Y}_{t+\tau}[\bar{\mathbf{a}}(t, t+\tau-1)]|\bar{\mathbf{H}}_t)$, for any intended future treatment assignment $\bar{\mathbf{a}}(t, t+\tau-1)$.

**Balancing representations.** The history $\bar{\mathbf{H}}_t = (\bar{\mathbf{X}}_t, \bar{\mathbf{A}}_{t-1}, \mathbf{V})$ of the patient contains the time-varying confounders $\bar{\mathbf{X}}_t$ which bias the treatment assignment $\mathbf{A}_t \in \{A_1, \ldots A_K\}$ in the observational dataset. Inverse probability of treatment weighting, as performed by MSMs, creates a pseudo-population where the probability of treatment $\mathbf{A}_t$ does not depend on the time-varying confounders (Robins et al., 2000). In this paper, we propose instead building a representation of the history $\bar{\mathbf{H}}_t$ that is not predictive of the treatment $\mathbf{A}_t$. This way, we remove the association between history, containing the time-varying confounders $\bar{\mathbf{X}}_t$, and current treatment $\mathbf{A}_t$. Robins (1999) shows that in this case, the estimation of counterfactual treatment outcomes is unbiased. See Appendix C for details and for an example of a causal graph with time-dependent confounders.

Let $\Phi$ be the representation function that maps the patient history $\bar{\mathbf{H}}_t$ to a representation space $\mathcal{R}$. To obtain unbiased treatment effects, $\Phi$ needs to construct treatment invariant representations such that $P(\Phi(\bar{\mathbf{H}}_t) \mid \mathbf{A}_t = A_1) = \cdots = P(\Phi(\bar{\mathbf{H}}_t) \mid \mathbf{A}_t = A_K)$. To achieve this and to estimate counterfactual outcomes under a planned sequence of treatments, we integrate the domain adversarial training framework proposed by Ganin et al. (2016) and extended by Sebag et al. (2019) to the multi-domain learning setting, into a sequence-to-sequence architecture. In our case, the different treatments at each timestep are considered the different domains. Note that the novelty here comes from the use of domain adversarial training to handle the bias from the time-dependent confounders, rather than the use of sequence-to-sequence models, which have already been applied to forecast treatment responses (Lim et al., 2018). Figure 2 illustrates our model architecture.

**Encoder.** The encoder network uses an RNN, with LSTM unit (Hochreiter & Schmidhuber, 1997), to process the history of treatments $\bar{\mathbf{A}}_{t-1}$, covariates $\bar{\mathbf{X}}_t$ and baseline features $\mathbf{V}$ to build a treatment invariant representation $\Phi(\bar{\mathbf{H}}_t)$, but also to predict one-step-ahead outcomes $\mathbf{Y}_{t+1}$. To achieve this, the encoder network aims to maximize the loss of the treatment classifier $G_a$ and minimize the loss

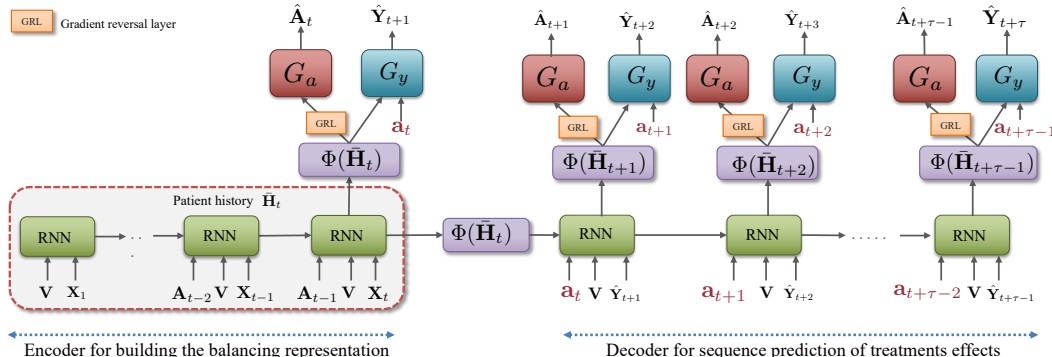

Figure 2: CRN architecture. Encoder builds representation $\Phi(\bar{\mathbf{H}}_t)$ that maximizes loss of treatment classifier $G_a$ and minimizes loss of outcome predictor $G_y$. $\Phi(\bar{\mathbf{H}}_t)$ is used to initialize the decoder, which continues to update it to predict counterfactual outcomes of a sequence of future treatments.

of the outcome predictor network $G_y$. This way, the balanced representation $\Phi(\bar{\mathbf{H}}_t)$ is not predictive of the assigned treatment $\mathbf{A}_t$, but is discriminative enough to estimate the outcome $\mathbf{Y}_{t+1}$. To train this model using gradient descent, we use the Gradient Reversal Layer (Ganin et al., 2016).

**Decoder.** The decoder network uses the balanced representation computed by the encoder to initialize the state of an RNN that predicts the counterfactual outcomes for a sequence of future treatments. During training, the decoder uses as input the outcomes from the observational data $(\mathbf{Y}_{t+1}, \ldots \mathbf{Y}_{t+\tau-1})$, the static patient features $\mathbf{V}$ and the intended sequence of treatments $\bar{\mathbf{a}}(t, t + \tau - 1)$. The decoder is trained in a similar way to the encoder to update the balanced representation and to estimate the outcomes. During testing, we do not have access to ground-truth outcomes; thus, the outcomes predicted by the decoder $(\hat{\mathbf{Y}}_{t+1}, \ldots \hat{\mathbf{Y}}_{t+\tau-1})$ are auto-regressively used instead as inputs. By running the decoder with different treatment settings, and by auto-regressively feeding back the outcomes, we can determine when to start and end different treatments, which is the optimal time to give the treatment and which treatments to give over time to obtain the best patient outcomes.

The representation $\Phi(\bar{\mathbf{H}}_t)$ is built by applying a fully connected layer, with Exponential Linear Unit (ELU) activation to the output of the LSTM. The treatment classifier $G_a$ and the predictor network $G_y$ consist of a hidden layer each, also with ELU activation. The output layer of $G_a$ uses softmax activation, while the output layer of $G_y$ uses linear activation for continuous predictions. For categorical outcomes, softmax activation can be used. We follow an approach similar to Lim et al. (2018) and we split the encoder and decoder training into separate steps. See Appendix E for details.

The encoder and decoder networks use variational dropout (Gal & Ghahramani, 2016) such that the CRN can also give uncertainty intervals for the treatment outcomes. This is particularity important in the estimation of treatment effects, since the model predictions should only be used when they have high confidence. Our model can also be modified to allow for irregular samplings of observations by using a PhasedLSTM (Neil et al., 2016).

## 5  ADVERSARIALLY BALANCED REPRESENTATION OVER TIME

At each timestep $t$, let the $K$ different possible treatments $\mathbf{A}_t \in \{A_1, \ldots A_K\}$ represent our domains. As described in Section 4, to remove the bias from time-dependent confounders, we build a representation of history $\bar{\mathbf{H}}_t$ that is invariant across treatments: $P(\Phi(\bar{\mathbf{H}}_t) \mid A_1) = \cdots = P(\Phi(\bar{\mathbf{H}}_t) \mid A_K)$.

This requirement can be enforced by minimizing the distance in the distribution of $\Phi(\bar{\mathbf{H}}_t)$ between any two pairs of treatments. Kifer et al. (2004); Ben-David et al. (2007), propose measuring the disparity between distributions based on their separability by a discriminatively-trained classifier. Let the symmetric hypothesis class $\mathcal{H}$ consist of the set of symmetric multiclass classifiers, such as neural network architectures. The $\mathcal{H}$-divergence between all pairs of two distributions is defined in terms of the capacity of the hypothesis class $\mathcal{H}$ to discriminate between examples from the multiple distributions. Empirically, minimizing the $\mathcal{H}-$divergence involves building a representation where examples from the multiple domains are as indistinguishable as possible (Ben-David et al., 2007; Li et al., 2018; Sebag et al., 2019). Ganin et al. (2016) use this idea to propose an adversarial framework

for domain adaptation involving building a representation which achieves maximum error on a domain classifier and minimum error on an outcome predictor. Similarly, in our case, we use domain adversarial training to build a representation of the patient history $\Phi(\bar{\mathbf{H}}_t)$ that is both invariant to the treatment given at timestep $t$, $\mathbf{A}_t$ and that achieves low error in estimating the outcome $\mathbf{Y}_{t+1}$.

Let $G_a(\Phi(\bar{\mathbf{H}}_t); \theta_a)$ be the treatment classifier with parameters $\theta_a$ and let $G_a^j(\Phi(\bar{\mathbf{H}}_t); \theta_a)$ be the output corresponding to treatment $A_j$. Let $G_y(\Phi(\bar{\mathbf{H}}_t); \theta_y)$ be the predictor network with parameters $\theta_y$. The representation function $\Phi$ is parameterized by the parameters $\theta_r$ in the RNN: $\Phi(\bar{\mathbf{H}}_t; \theta_r)$. Figure 3 shows the adversarial training procedure used.

For timestep $t$ and patient $(i)$, let $\mathcal{L}_{t,a}^{(i)}(\theta_r, \theta_a)$ be the treatment (domain) loss and let $\mathcal{L}_{t,y}^{(i)}(\theta_r, \theta_y)$ the outcome loss, defined as follows:

$$\mathcal{L}_{t,a}^{(i)}(\theta_r, \theta_a) = -\sum_{j=1}^{K} \mathbb{I}_{\{\mathbf{a}_t^{(i)} = a_j\}} \log(G_a^j(\Phi(\bar{\mathbf{H}}_t; \theta_r); \theta_a))$$

$$(2)$$

$$\mathcal{L}_{t,y}^{(i)}(\theta_r, \theta_y) = \|\mathbf{Y}_{t+1}^{(i)} - (G_y(\Phi(\bar{\mathbf{H}}_t; \theta_r), \theta_y))\|^2. \quad (3)$$

If the outcome is binary, the cross-entropy loss can be used instead for $\mathcal{L}_{t,y}$. To build treatment invariant representations and to also estimate patient outcomes, we aim to maximize treatment loss and minimize outcome loss.

Figure 3: Training procedure for building balancing representation.

Thus, the overall loss $\mathcal{L}_{t,y}^{(i)}$ at timestep $t$ is given by:

$$\mathcal{L}_t^{(i)}(\theta_r, \theta_y, \theta_a) = \sum_{i=1}^{N} \mathcal{L}_{t,y}^{(i)}(\theta_r, \theta_y) - \lambda \mathcal{L}_{t,a}^{(i)}(\theta_r, \theta_a), \quad (4)$$

where the hyperparameter $\lambda$ controls this trade-off between domain discrimination and outcome prediction. We use the standard procedure for training domain adversarial networks from Ganin et al. (2016) and we start off with an initial value for $\lambda$ and use an exponentially increasing schedule during training. To train the model using backpropagation, we use the Gradient Reversal Layer (GRL) (Ganin et al., 2016). For more details about the training procedure, see Appendix E.

By using the objective $\mathcal{L}_t^{(i)}(\theta_r, \theta_y, \theta_a)$, we reach the saddle point $(\hat{\theta}_r, \hat{\theta}_y, \hat{\theta}_a)$ that achieves the equilibrium between domain discrimination and outcome estimation.

$$(\hat{\theta}_r, \hat{\theta}_y) = \arg\min_{\theta_r, \theta_y} \mathcal{L}_t^{(i)}(\theta_r, \theta_y, \hat{\theta}_a) \qquad \hat{\theta}_a = \arg\max_{\theta_a} \mathcal{L}_t^{(i)}(\hat{\theta}_r, \hat{\theta}_y, \theta_a). \quad (5)$$

The result stated in Theorem 1 proves that the treatment (domain) loss part of our objective (from equation 2) aims to remove the time-dependent confounding bias.

**Theorem 1.** *Let $t \in \{1, 2, \dots\}$. For each $j = 1, ..., K$, let $P_j$ denote the distribution of $\bar{\mathbf{H}}_t$ conditional on $\mathbf{A}_t = A_j$ and let $P_j^{\Phi}$ denote the distribution of $\Phi(\bar{\mathbf{H}}_t)$ conditional on $\mathbf{A}_t = A_j$. Let $G_a^j$ denote the output of $G_a$ corresponding to treatment $A_j$. Then the minimax game defined by*

$$\min_{\Phi} \max_{G_a} \sum_{j=1}^{K} \mathbb{E}_{\bar{\mathbf{H}}_t \sim P_j} \left[ \log(G_a^j(\Phi(\bar{\mathbf{H}}_t); \theta_a)) \right] \qquad \text{subject to} \quad \sum_{j=1}^{K} G_a^j(\Phi(\bar{\mathbf{H}}_t)) = 1 \quad (6)$$

*has a global minimum which is attained if and only if $P_1^{\Phi} = P_2^{\Phi} = ... = P_K^{\Phi}$, i.e. when the learned representations are invariant across all treatments.*

*Proof.* This result is a restatement of the one in Li et al. (2018). For details, see the Appendix D. □

A good representation allows us to obtain a low error in estimating counterfactuals for all treatments, while at the same time to minimize the $\mathcal{H}$-divergence between induced marginal distributions of all the domains. We use an algorithm that directly minimizes a combination of the $\mathcal{H}-$divergence and the empirical training margin.

## 6 EXPERIMENTS

In real datasets, counterfactual outcomes and the degree of time-dependent confounding are not known (Schulam & Saria, 2017; Lim et al., 2018). To validate the CRN[1], we evaluate it on a Pharmacokinetic-Pharmacodynamic model of tumour growth (Geng et al., 2017), which uses a state-of-the-art bio-mathematical model to simulate the combined effects of chemotherapy and radiotherapy in lung cancer patients. The same model was used by Lim et al. (2018) to evaluate RMSNs.

**Model of tumour growth** The volume of tumour $t$ days after diagnosis is modelled as follows:

$$V(t+1) = \Big(1 + \underbrace{\rho\log\big(\frac{K}{V(t)}\big)}_{\text{Tumor growth}} - \underbrace{\beta_c C(t)}_{\text{Chemotherapy}} - \underbrace{(\alpha_r d(t) + \beta_r d(t)^2)}_{\text{Radiotherapy}} + \underbrace{e_t}_{\text{Noise}}\Big)V(t) \tag{7}$$

where $K, \rho, \beta_c, \alpha_r, \beta_r, e_t$ are sampled as described in Geng et al. (2017). To incorporate heterogeneity in patient responses, the prior means for $\beta_c$ and $\alpha_r$ are adjusted to create patient subgroups, which are used as baseline features. The chemotherapy concentration $C(t)$ and radiotherapy dose $d(t)$ are modelled as described in Appendix F. Time-varying confounding is introduced by modelling chemotherapy and radiotherapy assignment as Bernoulli random variables, with probabilities $p_c$ and $p_r$ depending on the tumour diameter: $p_c(t) = \sigma\big(\frac{\gamma_c}{D_{\max}}(\bar{D}(t) - \delta_c)\big)$ and $p_r(t) = \sigma\big(\frac{\gamma_r}{D_{\max}}(\bar{D}(t) - \delta_r)\big)$ where $\bar{D}(t)$ is the average diameter over the last 15 days, $D_{\max} = 13$cm, $\sigma(\cdot)$ is the sigmoid and $\delta_c = \delta_r = D_{\max}/2$. The amount of time-dependent confounding is controlled through $\gamma_c, \gamma_r$; the higher $\gamma_\star$ is, the more important the history is in assigning treatments. At each timestep, there are four treatment options: no treatment, chemotherapy, radiotherapy, combined chemotherapy and radiotherapy. For details about data simulation, see Appendix F.

**Benchmarks** We used the following benchmarks for performance comparison: Marginal Structural Models (MSMs) (Robins et al., 2000), which use logistic regression for estimating the IPTWs and linear regression for prediction (see Appendix G for details). We also compare against the Recurrent Marginal Structural Networks (RMSNs) Lim et al. (2018), which is the current state-of-the-art model in estimating treatment responses. RMSNs use RNNs to estimate the IPTWs and the patient outcomes (details in Appendix H). To show that standard supervised learning models do not handle the time-varying confounders we compare against an RNN and a linear regression model, which receive as input treatments and covariates to predict the outcome (see Appendix I for details). Our model architecture follows the description in Sections 4 and 5, with full training details and hyperparameter optimization in Appendix J. To show the importance of adversarial training, we also benchmark against CRN ($\lambda = 0$) a model with the same architecture, but with $\lambda = 0$, i.e our model architecture without adversarial training.

### 6.1 EVALUATE MODELS ON COUNTERFACTUAL PREDICTIONS

Previous methods focused on evaluating the error only for factual outcomes (observed patient outcomes) (Lim et al., 2018). However, to build decision support systems, we need to evaluate how well the models estimate the counterfactual outcomes, i.e patient outcomes under alternative treatment options. The parameters $\gamma_c$ and $\gamma_r$ control the treatment assignment policy, i.e. the degree of time-dependent confounding present in the data. We evaluate the benchmarks under different degrees of time-dependent confounding by setting $\gamma = \gamma_c = \gamma_r$. For each $\gamma$ we simulate a 10000 patients for training, 1000 for validation (hyperparameter tuning) and 1000 for out-of-sample testing. For the patients in the test set, for each time $t$, we also simulate counterfactuals $\mathbf{Y}_{t+1}$, represented by tumour volume $V(t+1)$, under all possible treatment options.

---

[1]The implementation of the model can be found at `https://bitbucket.org/mvdschaar/mlforhealthlabpub/src/master/alg/counterfactual_recurrent_network/` and at `https://github.com/ioanabica/Counterfactual-Recurrent-Network`.

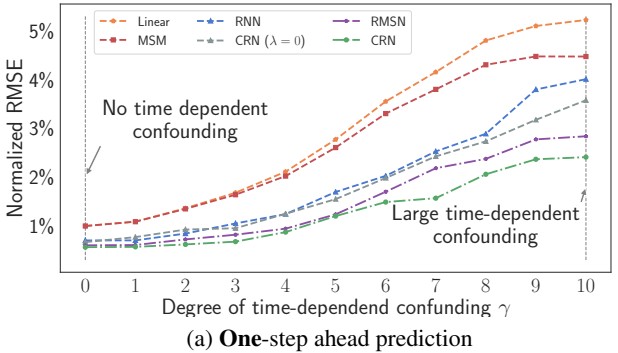
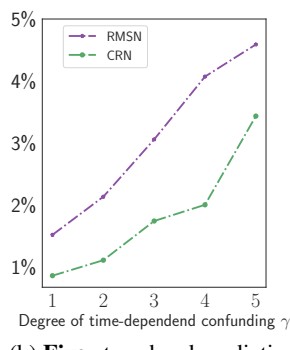

(a) **One**-step ahead prediction

(b) **Five**-step ahead prediction

Figure 4: Results for prediction of patient counterfactuals.

Figure 4 (a) shows the normalized root mean squared error (RMSE) for one-step ahead estimation of counterfactuals with varying degree of time-dependent confounding $\gamma$. The RMSE is normalized by the maximum tumour volume: $V_{max} = 1150\text{cm}^3$. The linear and MSM models provide a baseline for performance as they achieve the highest RMSE. While the use of IPTW in MSMs helps when $\gamma$ increases, using linear modelling has severe limitations. When there is no time-dependent confounding, the machine learning methods achieve similar performance, close to 0.6% RMSE. As the bias in the dataset increases, the harder it becomes for the RNN and the CRN ($\lambda = 0$) to generalize to estimate outcomes of treatments not matching the training policy. When $\gamma = 10$, CRN improves by 48.1% on the same model architecture without domain adversarial training CRN ($\lambda = 0$).

Our proposed model achieves the lowest RMSE across all values of $\gamma$. Compared to RMSNs, CRN improves by $\sim 17\%$ when $\gamma > 6$. To highlight the gains of our method even for smaller $\gamma$, Figure 4 (b) shows the RMSE for five-step ahead prediction (with counterfactuals generated as described in Section 6.2 and Appendix L). RMSNs also use a decoder for sequence prediction. However, RMSNs require training additional RNNs to estimate the IPTW, which are used to weight each sample during the decoder training. For $\tau$-step ahead prediction, IPTW involves multiplying $\tau$ weights which can result in high variance. The results in Figure 4 (b) show the problems with using IPTW to handle the time-dependent confounding bias. See Appendix K for more results on multi-step ahead prediction.

**Balancing representation:** To evaluate whether the CRN has indeed learnt treatment invariant represenations, for $\gamma = 5$, we illustrate in Figure 5 the T-SNE embeddings of the balancing rep-resentations $\Phi(\bar{\mathbf{H}}_t)$ built by the CRN encoder for test patients. We color each point by the treat-ment $\mathbf{A}_t \in \{\text{no treatment}, \text{chemotherapy}, \text{radiotherapy}, \text{combined chemotherapy and radiotherapy}\}$ received at timestep $t$ to highlight the invariance of $\Phi(\bar{\mathbf{H}}_t)$ across the different treatments. In Figure 5(b), we show $\Phi(\bar{\mathbf{H}}_t)$ only for chemotherapy and radiotherapy for better understanding.

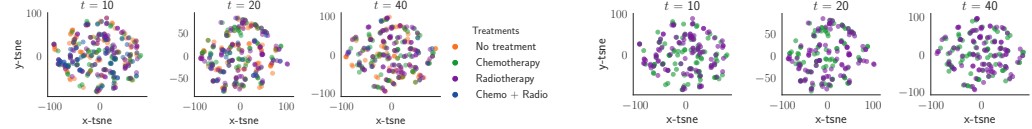

Figure 5: TSNE embedding of the balancing representation $\Phi(\bar{\mathbf{H}}_t)$ learnt by the CRN encoder at different timesteps $t$. Notice that $\Phi(\bar{\mathbf{H}}_t)$ is not predictive of the treatment $\mathbf{A}_t$ given at timestep $t$.

## 6.2 EVALUATE RECOMMENDING THE RIGHT TREATMENT AND TIMING OF TREATMENT

Evaluating the models just in terms of the RMSE on counterfactual estimation is also not enough for assessing their reliability when used as part of decision support systems. In this section we assess how well the models can select the correct treatment and timing of treatment for several forecasting horizons $\tau$. We generate test sets consisting of 1000 patients where for each horizon $\tau$ and for each time $t$ in a patient's trajectory, there are $\tau$ options for giving chemotherapy at one of $t, \ldots t + \tau - 1$ and $\tau$ options for giving radiotherapy at one of $t, \ldots t + \tau - 1$. At the rest of the future timesteps, no treatment is applied. These $2\tau$ treatment plans are assessed in terms of the tumour volume outcome $\mathbf{Y}_{t+\tau}$. We select the treatment (chemotherapy or radiotherapy) that achieves lowest $\mathbf{Y}_{t+\tau}$, and within the correct treatment the timing with lowest $\mathbf{Y}_{t+\tau}$. We also compute the normalized RMSE for

Table 1: Results for recommending the correct treatment and timing of treatment.

| | $\tau$ | $\gamma_c = 5, \gamma_r = 5$ | | | $\gamma_c = 5, \gamma_r = 0$ | | | $\gamma_c = 0, \gamma_r = 5$ | | |
| --- | --- | --- | --- | --- | --- | --- | --- | --- | --- | --- |
| | | CRN | RMSN | MSM | CRN | RMSN | MSM | CRN | RMSN | MSM |
| Normalized RMSE | 3 | **2.43%** | 3.16% | 6.75% | **1.08%** | 1.35% | 3.68% | **1.54%** | 1.59% | 3.23% |
| | 4 | **2.83%** | 3.95% | 7.65% | **1.21%** | 1.81% | 3.84% | **1.81%** | 2.25% | 3.52% |
| | 5 | **3.18%** | 4.37% | 7.95% | **1.33%** | 2.13% | 3.91% | **2.03%** | 2.71% | 3.63% |
| | 6 | **3.51%** | 5.61% | 8.19% | **1.42%** | 2.41% | 3.97% | **2.23%** | 2.73% | 3.71% |
| | 7 | **3.93%** | 6.21% | 8.52% | **1.53%** | 2.43% | 4.04% | **2.43%** | 2.88% | 3.79% |
| Treatment Accuracy | 3 | **83.1%** | 75.3% | 73.9% | **83.2%** | 78.6% | 77.1% | **92.9%** | 87.3% | 74.9% |
| | 4 | **82.5%** | 74.1% | 68.5% | **81.3%** | 77.7% | 73.9% | **85.7%** | 83.8% | 74.1% |
| | 5 | **73.5%** | 72.7% | 63.2% | **78.3%** | 77.2% | 72.3% | **83.8%** | 82.1% | 72.8% |
| | 6 | **69.4%** | 66.7% | 62.7% | **79.5%** | 76.3% | 71.8% | **78.6%** | 69.7% | 64.5% |
| | 7 | **71.2%** | 68.8% | 62.4% | **72.7%** | 71.8% | 71.6% | **71.9%** | 69.3% | 61.2% |
| Treatment Timing Accuracy | 3 | **79.6%** | 78.1% | 67.6% | **80.5%** | 76.8% | 77.5% | **79.8%** | 75.7% | 60.6% |
| | 4 | **73.9%** | 70.3% | 63.1% | **79.0%** | 77.2% | 73.4% | **75.4%** | 71.4% | 58.2% |
| | 5 | **69.8%** | 68.6% | 62.4% | **78.3%** | 73.3% | 63.6% | **66.9%** | 31.3% | 29.5% |
| | 6 | **66.9%** | 66.2% | 62.6% | **73.5%** | 72.1% | 63.9% | **65.8%** | 24.2% | 15.5% |
| | 7 | **64.5%** | 63.6% | 62.2% | **70.6%** | 57.4% | 44.2% | **63.9%** | 25.6% | 12.5% |

predicting $Y_{t+\tau}$. See Appendix L for more details about the test set. The models are evaluated for 3 settings of $\gamma_c$ and $\gamma_r$.

Table 1 shows the results for this evaluation set-up. The treatment accuracy denotes the percentage of patients for which the correct treatment was selected, while the treatment timing accuracy is the percentage for which the correct timing was selected. Note that when $\gamma_c = 0$ and $\gamma_r = 5$, RMSN and MSM select the wrong treatment timing for projection horizons $\tau > 4$. CRN performs similarly among the different policies present in the observational data and achieve the lowest RMSE and highest accuracy in selecting the correct treatment and timing of treatment.

In Appendix M we also show the applicability of the CRN in more complex medical scenarios involving real data. We provide experimental results based on the Medical Information Mart for Intensive Care (MIMIC III) database (Johnson et al., 2016) consisting of electronic health records from patients in the ICU.

# 7 CONCLUSION

Despite its wide applicability, the problem of causal inference for time-dependent treatments has been relatively less studied compared to problem of causal inference in the static setting. Both new methods and theory are necessary to be able to harness the full potential of observational data for learning individualized effects of complex treatment scenarios. Further work in this direction is needed for proposing alternative methods for handling time-dependent confounders, for modelling combinations of treatments assigned over time or for estimating the individualized effects of time-dependent treatments with associated dosage.

In this paper, we introduced the Counterfactual Recurrent Network (CRN), a model that estimates individualized effects of treatments over time using a novel way of handling the bias from time-dependent confounders through adversarial training. Using a model of tumour growth, we validated CRN in realistic medical scenarios and we showed improvements over existing state-of-the-art methods. We also showed the applicability of the CRN a real dataset consiting of patient electronic health records. The counterfactual predictions of CRN have the potential to be used as part of clinical decision support systems to address relevant medical challenges involving selecting the best treatments for patients over time, identify optimal treatment timings but also when the treatment is no longer needed. In future work, we will aim to build better balancing representations and to provide theoretical guarantees for the expected error on the counterfactuals.

ACKNOWLEDGMENTS

We would like to thank the reviewers for their valuable feedback. The research presented in this paper was supported by The Alan Turing Institute, under the EPSRC grant EP/N510129/1 and by the US Office of Naval Research (ONR).

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

Appendix

# A    Extended related work

**Causal inference in the static setting:** A large number of methods have been proposed to learn treatment effects from observational data in the static setting. In this case, it is needed to adjust for the selection bias; bias caused by the fact that, in the observational dataset, the treatment assignments depend on the patient features. Several ways of handling the selection bias involve using propensity matching (Austin, 2011; Imai & Ratkovic, 2014; Abadie & Imbens, 2016), building representations where treated and un-treated populations had similar distributions (Johansson et al., 2016; Shalit et al., 2017; Li & Fu, 2017; Yao et al., 2018) or performing propensity-aware hyperparameter tuning (Alaa & van der Schaar, 2017; 2018). However, these methods for the static setting cannot be extended directly to time-varying treatments (Hernán et al., 2000; Schisterman et al., 2009).

**Learning optimal policies:** A related problem to ours involves learning the optimal treatment policies from logged data (Swaminathan & Joachims, 2015a;b; Atan et al., 2018). That is, learning the treatment option that would give the best reward. Note the difference to the causal inference setting considered in this paper, where the aim is to learn the counterfactual patient outcomes under all possible treatment options. Learning all of the counterfactual outcomes is a harder problem and can also be used for finding the optimal treatment.

A method for learning optimal policies, proposed by Atan et al. (2018) uses domain adversarial training to build a representation that is invariant to the following two domains: observational data and simulated randomized clinical trial data, where the treatments have equal probabilities. Atan et al. (2018) only considers the static setting and aims to choose the optimal treatment instead of estimating all of the counterfactual outcomes. In our paper the aim is to eliminate the bias from the time-dependent confounders and reliably estimate *all of the potential outcomes*; thus, at each timestep $t$ we build a representation that is invariant to the treatment.

**Off-policy evaluation in reinforcement learning:** In reinforcement learning, a similar problem to ours is off-policy evaluation, which uses retrospective observational data, also known as logged bandit feedback (Hoiles & Van Der Schaar, 2016; Păduraru et al., 2013; Doroudi et al., 2017). In this case, the retrospective observational data consists of sequences of states, actions and rewards which were generated by an agent operating under an unknown policy. The off-policy evaluation methods aim to use this data to estimate the expected reward of a target policy. These methods use algorithms based on importance sampling (Precup, 2000; Thomas et al., 2015; Guo et al., 2017), action-value function approximation (model based) (Hallak et al., 2015) or doubly robust combination of both approaches (Jiang & Li, 2015). Nevertheless, these methods focus on obtaining average rewards of policies, while in our case the aim is to estimate individualized patient outcomes for future treatments.

# B    Assumptions

The standard assumptions needed for identifying the treatment effects are (Robins & Hernán, 2008; Lim et al., 2018; Schulam & Saria, 2017):

**Assumption 1: Consistency**. If $\mathbf{A}_t = \mathbf{a}_t$ for a given patient, then the potential outcome for treatment $\mathbf{a}_t$ is the same as the observed (factual) outcome: $\mathbf{Y}_{t+1}[\mathbf{a}_t] = \mathbf{Y}_{t+1}$.

**Assumption 2: Positivity (Overlap)** (Imai & Van Dyk, 2004): If $P(\bar{\mathbf{A}}_{t-1} = \bar{\mathbf{a}}_{t-1}, \bar{\mathbf{X}}_t = \bar{\mathbf{x}}_t) \neq 0$ then $P(\mathbf{A}_t = \mathbf{a}_t \mid \bar{\mathbf{A}}_{t-1} = \bar{\mathbf{a}}_{t-1}, \bar{\mathbf{X}}_t = \bar{\mathbf{x}}_t) > 0$ for all $\bar{\mathbf{a}}_t$.

**Assumption 3: Sequential strong ignorability.** $\mathbf{Y}_{t+1}[\mathbf{a}_t] \perp\!\!\!\perp \mathbf{A}_t \mid \bar{\mathbf{A}}_{t-1}, \bar{\mathbf{X}}_t, \forall \mathbf{a}_t \in \mathcal{A}, \forall t$.

Assumption 2 means that, for each timestep, each treatment has non-zero probability of being assigned. Assumption 3 means that there are no hidden confounders, that is, all of covariates affecting both the treatment assignment and the outcomes are present in the the observational dataset. Note that while assumption 3 is standard across all methods for estimating treatment effects, it is not testable in practice (Robins et al., 2000; Pearl et al., 2009).

## C TIME-DEPENDENT CONFOUNDING

Figure 6 illustrates the causal graphs for a time-varying exposures with 2-steps (Robins et al., 2000). In Figure 6 (a), the covariate $X$ is a time-dependent confounder because it affects the treatment assignments and at the same time, its value is changed by past treatments (Mansournia et al., 2017), as illustrated by the red arrows. Thus, the treatment probabilities at each time $t$ depend on the history of covariate $X$ and past treatments. Note that $U_0$ and $U_1$ are hidden variables which only affect the covariates, i.e. they do not have arrows into the treatments. Thus, the no hidden confounders assumption (Assumption 3) is satisfied.

Figure 6 (a) and (b) illustrate the two cases when there is no bias from time-dependent confounding. In Figure 6 (a) the treatment probabilities are independent, while in Figure 6 (b) they depend on past treatments.

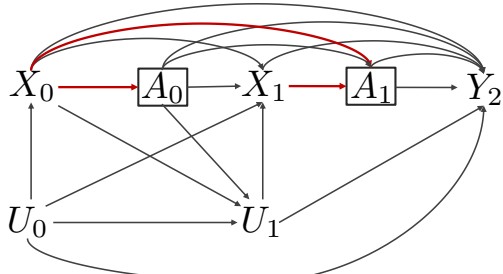

(a) Bias from time-dependent confounders.

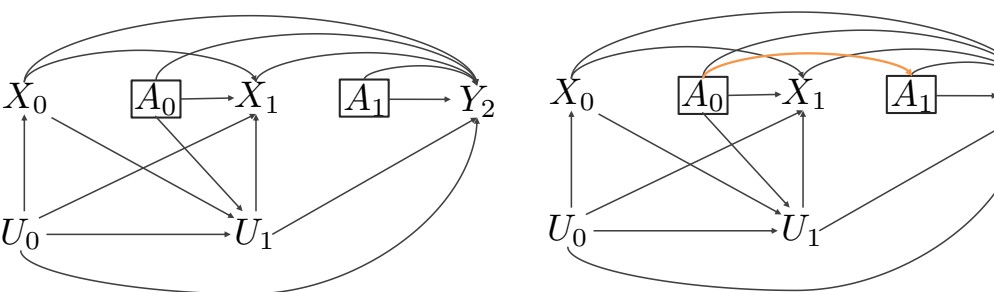

(a) Option 1 for removing bias from time-dependent confounders.

(a) Option 2 for removing bias from time-dependent confounders.

Figure 6: Causal graphs for 2-step time-varying exposures (Robins et al., 2000). $X_0, X_1$ are patient covariates, $A_0, A_1$ are treatments, $U_0, U_1$ are unobserved variable and $Y_2$ is the outcome.

**Marginal Structural Models** Robins et al. (2000). To remove the association between time-dependent confounders and time-varying treatments, Marginal Structural Models propose using inverse probability of treatment weighting (IPTW). Without loss of generality, consider the use of MSMs with univariate treatments, baseline variables and outcomes. The outcome after $t$ timesteps is parametrized as follows: $\mathbf{E}[Y_{t+1} \mid \mathbf{a}_1, \ldots \mathbf{a}_t, V] = g(\mathbf{a}_1, \ldots \mathbf{a}_n, V; \theta)$, where $g(\cdot)$ is usually a linear function with parameters $\theta$. To remove the bias from the time-dependent confounders present in the observational dataset, in the regression model $g(\cdot)$ MSMs weights each patients using either stabilized weights:

$$SW(t) = \prod_{l=1}^{t} \frac{f(\mathbf{A}_l \mid \bar{\mathbf{A}}_{l-1})}{f(\mathbf{A}_l \mid \bar{\mathbf{X}}_l, \bar{\mathbf{A}}_{l-1}, \mathbf{V})} \tag{8}$$

or unstabilized weights:

$$W(t) = \prod_{l=1}^{t} \frac{1}{f(\mathbf{A}_l \mid \bar{\mathbf{X}}_l, \bar{\mathbf{A}}_{l-1}, \mathbf{V})}, \tag{9}$$

where $f(\cdot)$ represents the conditional probability mass function for discrete treatments.

Inverse probability of treatment weighting (IPTW) creates a pseudo-population where each member consists of themselves and $W - 1$ (or $SW - 1$) copies added though weighting. In this pseudo-population, Robins Robins (1999) shows that $\bar{\mathbf{X}}_t$ does not predict treatment $\mathbf{A}_t$, thus removing the bias from time-dependent confounders.

When using unstabilized weights $W$, the causal graph in the pseudo-population is the one in Figure 6 (a) where $P(\mathbf{A}_t \mid \bar{\mathbf{X}}_t, \bar{\mathbf{A}}_{t-1}, V) = P(\mathbf{A}_t)$. On the other hand, when using stabilized weights $SW$, causal graph in the pseudo-population is the one in Figure 6 (b) where $P(\mathbf{A}_t \mid \bar{\mathbf{X}}_t, \bar{\mathbf{A}}_{t-1}, V) = P(\mathbf{A}_t \mid \bar{\mathbf{A}}_{t-1})$.

**Counterfactual Recurrent Networks**. Instead of using IPTW, we proposed building a representation of $\bar{\mathbf{X}}_t, \bar{\mathbf{A}}_{t-1}, V$ that is not predictive of treatment $\mathbf{A}_t$. At timestep $t$, we have $k$ different possible treatments $\mathbf{A}_t \in \{A_1, \dots A_K\}$. We build a representation of the history and covariates and treatments that has the same distribution across the different possible treatments: $P(\Phi(\bar{\mathbf{X}}_t, \bar{\mathbf{A}}_{t-1}, \mathbf{V}) \mid \mathbf{A}_t = A_1) = \cdots = P(\Phi(\bar{\mathbf{X}}_t, \bar{\mathbf{A}}_{t-1}, \mathbf{V}) \mid \mathbf{A}_t = A_K)$. By breaking the association between past exposure and current treatments $\mathbf{A}_t$, we satisfy the causal graph in Figure 6 (a) and thus we remove the bias from time-dependent confounders.

## D   PROOF OF THEOREM 1

We first prove the following proposition.

**Proposition 1.** *For fixed $\Phi$, let $x' = \Phi(\bar{\mathbf{h}}_t)$. Then the optimal prediction probabilities of $G_a$ are given by*

$$G_a^{j\,*}(x') = \frac{P_j^{\Phi}(x')}{\sum_{i=1}^{K} P_i^{\Phi}(x')} \,. \tag{10}$$

*Proof.* For fixed $\Phi$, the optimal prediction probabilities are given by

$$G_a^* = \arg\max_{G_a} \sum_{j=1}^{K} \int_{x'} \log(G_a^j(x')) P_j^{\Phi}(x') dx' \qquad \text{subject to} \sum_{j=1}^{K} G_a^j(x') = 1 \,. \tag{11}$$

Maximising the value function pointwise and applying Lagrange multiplies, we get

$$G_a^* = \arg\max_{G_a} \sum_{j=1}^{K} \log(G_a^j(x')) P_j^{\Phi}(x') + \lambda \left( \sum_{j=1}^{K} G_a^j(x') - 1 \right) . \tag{12}$$

Setting the derivative (w.r.t. $G_a^{j\,*}(x')$) to 0 and solving for $G_a^{j\,*}(x')$ we get

$$G_a^{j\,*}(x') = -\frac{P_j^{\Phi}(x')}{\lambda} \tag{13}$$

where $\lambda$ can now be solved for using the constraint to be $\lambda = -\sum_{i=1}^{K} P_i^{\Phi}(x')$. This gives the result. $\qquad\square$

*Proof.* (of **Theorem 1**) By substituting the expression from Proposition 1 into the minimax game defined in Eq. 6, the objective for $\Phi$ becomes

$$\min_{\Phi} \sum_{j=1}^{K} \mathbb{E}_{x' \sim P_j^{\Phi}} \left[ \log \left( \frac{P_j^{\Phi}(x')}{\sum_{i=1}^{K} P_i^{\Phi}(x')} \right) \right] . \tag{14}$$

We then note that

$$\sum_{j=1}^{K} \mathbb{E}_{x' \sim P_j^{\Phi}} \left[ \log \left( \frac{P_j^{\Phi}(x')}{\sum_{i=1}^{K} P_i^{\Phi}(x')} \right) \right] + K \log K = \sum_{j=1}^{K} \left( \mathbb{E}_{x' \sim P_j^{\Phi}} \left[ \log \left( \frac{P_j^{\Phi}(x')}{\sum_{i=1}^{K} P_i^{\Phi}(x')} \right) \right] + \log K \right) \tag{15}$$

$$= \sum_{j=1}^{K} \mathbb{E}_{x' \sim P_j^{\Phi}} \left[ \log \left( \frac{P_j^{\Phi}(x')}{\frac{1}{K} \sum_{i=1}^{K} P_i^{\Phi}(x')} \right) \right] \tag{16}$$

$$= \sum_{j=1}^{K} KL \left( P_j^{\Phi}(x') \middle\| \frac{1}{K} \sum_{i=1}^{K} P_i^{\Phi}(x') \right) \tag{17}$$

$$= K \cdot JSD(P_1^{\Phi}, ..., P_K^{\Phi}) \tag{18}$$

where $KL(\cdot||\cdot)$ is the Kullback-Leibler divergence and $JSD(\cdot, ..., \cdot)$ is the multi-distribution Jensen-Shannon Divergence (Li et al., 2018). Since $K \log K$ is a constant and the multi-distribution JSD is non-negative and 0 if and only if all distributions are equal, we have that $P_1^{\Phi} = ... = P_K^{\Phi}$. $\qquad\square$

# E    TRAINING PROCEDURE FOR CRN

Let $\mathcal{D} = \left\{ \{\mathbf{x}_t^{(i)}, \mathbf{a}_t^{(i)}, \mathbf{y}_{t+1}^{(i)}\}_{t=1}^{T^{(i)}} \cup \{\mathbf{v}^{(i)}\} \right\}_{i=1}^{N}$ be an observational dataset consisting of information about $N$ independent patients that we use to train CRN. The encoder and decoder networks part of CRN are trained into two separate steps.

To begin with, the encoder is trained to built treatment invariant representations of the patient history and to perform one-step ahead prediction. After the encoder is optimized, we use it to compute the balancing representation $\mathbf{br}_t^{(i)}$ for each timestep in the trajectory of patient $(i)$. To train the decoder, we modify the training dataset as follows. For each patient $(i)$, we split their trajectory into shorter sequences of the $\tau_{\max}$ timesteps of the form:

$$\left\{ \mathbf{br}_l^{(i)} \cup \{\mathbf{y}_{l+t}^{(i)}, \mathbf{a}_{l+t}^{(i)}, \mathbf{y}_{l+t+1}^{(i)}\}_{t=1}^{\tau_{max}} \cup \mathbf{v}^{(i)} \right\}, \tag{19}$$

for $l = 1, \ldots T^{(i)} - \tau_{\max}$. Thus, each patients contributes with $T^{(i)} - \tau_{\max}$ examples in the dataset for training the decoder. The different sequences obtained for all patents are randomly grouped into minibatches and used for training.

The pseudocode in Algorithm 1 shows the training procedure used for the encoder and decoder networks part of CRN. The model was implemented in TensorFlow and trained on an NVIDIA Tesla K80 GPU. The Adam optimizer (Kingma & Ba, 2014) was used for training and both the encoder and the decoder are trained for 100 epochs.

---

**Algorithm 1** Pseudo-code for training CRN

---

**Input:** Training data: $\mathcal{D} = \left\{ \{\mathbf{x}_t^{(i)}, \mathbf{a}_t^{(i)}, \mathbf{y}_{t+1}^{(i)}\}_{t=1}^{T^{(i)}} \cup \mathbf{v}^{(i)} \right\}_{i=1}^{N}$

**(1) Encoder optimization**: parameters $\theta_{E,r}, \theta_{E,a}, \theta_{E,y}$.
Learning rate: $\mu$
**for** $p = 1, \ldots, \text{max epochs}$ **do**
$\quad \lambda_p = \dfrac{2}{1 + \exp(-10 \cdot p)} - 1$
$\quad$ **for** Batch $\mathcal{B} = \left\{ \{\mathbf{x}_t^{(i)}, \mathbf{a}_t^{(i)}, \mathbf{y}_{t+1}^{(i)}\}_{t=0}^{T^{(i)}} \cup \mathbf{v}^{(i)} \right\}_{i=1}^{|\mathcal{B}|}$ in epoch **do**
$\quad\quad$ Compute $\mathcal{L}_{E,a}^{\mathcal{B}}(\theta_{E,r}, \theta_{E,a}) = \frac{1}{|\mathcal{B}|} \sum_{i \in \mathcal{B}} \sum_{t=1}^{T^{(i)}} \mathcal{L}_{t,a}^{(i)}(\theta_{E,r}, \theta_{E,a})$
$\quad\quad$ Compute $\mathcal{L}_{E,y}^{\mathcal{B}}(\theta_{E,r}, \theta_{E,y}) = \frac{1}{|\mathcal{B}|} \sum_{i \in \mathcal{B}} \sum_{t=1}^{T^{(i)}} \mathcal{L}_{t,y}^{(i)}(\theta_{E,r}, \theta_{E,y})$
$\quad\quad \theta_{E,r} \leftarrow \theta_{E,r} - \mu \left( \frac{\partial \mathcal{L}_{E,y}^{\mathcal{B}}(\theta_{E,r}, \theta_{E,y})}{\partial \theta_{E,r}} - \lambda_p \frac{\partial \mathcal{L}_{E,a}^{\mathcal{B}}(\theta_{E,r}, \theta_{E,a})}{\partial \theta_{E,r}} \right)$
$\quad\quad \theta_{E,y} \leftarrow \theta_{E,y} - \mu \frac{\partial \mathcal{L}_{E,y}^{\mathcal{B}}(\theta_{E,r}, \theta_{E,y})}{\partial \theta_{E,y}}$
$\quad\quad \theta_{E,a} \leftarrow \theta_{E,a} - \mu \frac{\partial \mathcal{L}_{E,a}^{\mathcal{B}}(\theta_{E,r}, \theta_{E,a})}{\partial \theta_{E,a}}$
$\quad$ **end for**
**end for**

**(2) Compute the encoder balanced representation and use it to initialize the decoder hidden state.**
**for** $i = 1, ..., N$ **do**
$\quad$ **for** $t = 1, \ldots, T^{(i)}$ **do**
$\quad\quad \mathbf{br}_t^{(i)} = \text{encoder}(\bar{\mathbf{x}}_t^{(i)}, \bar{\mathbf{a}}_{t-1}^{(i)}, \mathbf{v}^{(i)}; \theta_{E,r})$
$\quad$ **end for**
**end for**

**(3) Split dataset in sequences of $\tau_{\max}$ timesteps**:

$$\left\{ \left\{ \mathbf{br}_l^{(i)} \cup \{\mathbf{y}_{l+t}^{(i)}, \mathbf{a}_{l+t}^{(i)}, \mathbf{y}_{l+t+1}^{(i)}\}_{t=1}^{\tau_{max}} \cup \mathbf{v}^{(i)} \right\}_{l=1}^{T^{(i)} - \tau_{\max}} \right\}_{i=1}^{N}$$

**(4) Optimize decoder**: parameters $\theta_{D,r}, \theta_{D,a}, \theta_{D,y}$
Learning rate: $\mu$
**for** $p = 1, \ldots, \text{max epochs}$ **do**
$\quad \lambda_p = \dfrac{2}{1 + \exp(-10 \cdot p)} - 1$
$\quad$ **for** Batch $\mathcal{B} = \left\{ \mathbf{br}_l^{(i)} \cup \{\mathbf{y}_{l+t}^{(i)}, \mathbf{a}_{l+t}^{(i)}, \mathbf{y}_{l+t+1}^{(i)}\}_{t=0}^{\tau_{max}} \cup \{\mathbf{v}^{(i)}\} \right\}_{i=1}^{|\mathcal{B}|}$ in epoch **do**
$\quad\quad$ Compute $\mathcal{L}_{D,a}^{\mathcal{B}}(\theta_{D,r}, \theta_{D,a}) = \frac{1}{|\mathcal{B}|} \sum_{i \in \mathcal{B}} \sum_{t=1}^{\tau_{max}} \mathcal{L}_{t,a}^{(i)}(\theta_{D,r}, \theta_{D,a})$
$\quad\quad$ Compute $\mathcal{L}_{D,y}^{\mathcal{B}}(\theta_{D,r}, \theta_{D,y}) = \frac{1}{|\mathcal{B}|} \sum_{i \in \mathcal{B}} \sum_{t=1}^{\tau_{max}} \mathcal{L}_{t,y}^{(i)}(\theta_{D,r}, \theta_{D,y})$
$\quad\quad \theta_{D,r} \leftarrow \theta_{D,r} - \mu \left( \frac{\partial \mathcal{L}_{D,y}^{\mathcal{B}}(\theta_{D,r}, \theta_{D,y})}{\partial \theta_{D,r}} - \lambda_p \frac{\partial \mathcal{L}_{D,a}^{\mathcal{B}}(\theta_{D,r}, \theta_{D,a})}{\partial \theta_{D,r}} \right)$
$\quad\quad \theta_{D,y} \leftarrow \theta_{D,y} - \mu \frac{\partial \mathcal{L}_{D,y}^{\mathcal{B}}(\theta_{D,r}, \theta_{D,y})}{\partial \theta_{D,y}}$
$\quad\quad \theta_{D,a} \leftarrow \theta_{D,a} - \mu \frac{\partial \mathcal{L}_{D,a}^{\mathcal{B}}(\theta_{D,r}, \theta_{D,a})}{\partial \theta_{D,a}}$
$\quad$ **end for**
**end for**

**Output:** Trained CRN encoder (parameters $\theta_{E,r}, \theta_{E,a}, \theta_{E,y}$) and trained CRN decoder (parameters $\theta_{D,r}, \theta_{D,a}, \theta_{D,y}$.)

---

## F  PHARMACOKINETIC-PHARMACODYNAMIC MODEL OF TUMOUR GROWTH

To evaluate the CRN on counterfactual estimation, we need access to the data generation mechanism to build a test set that consists of patient outcomes under all possible treatment options. For this purpose, we use the state-of-the-art pharmacokinetic-pharmacodynamic (PK-PD) model of tumour growth proposed by Geng et al. (2017) and also used by Lim et al. (2018) for evaluating RMSMs. The PK-PD model characterizes patients suffering from non-small cell lung cancer and models the evolution of their tumour under the combined effects of chemotherapy and radiotherapy. In addition, the model includes different distributions of tumour sizes based on the cancer stage at diagnosis.

**Model of tumour growth** The volume of tumour $t$ days after diagnosis is modelled as follows:

$$V(t+1) = \left(1 + \underbrace{\rho\log(\frac{K}{V(t)})}_{\text{Tumor growth}} - \underbrace{\beta_c C(t)}_{\text{Chemotherapy}} - \underbrace{(\alpha_r d(t) + \beta_r d(t)^2)}_{\text{Radiotherapy}} + \underbrace{e_t}_{\text{Noise}}\right)V(t) \tag{20}$$

where the parameters $K, \rho, \beta_c, \alpha_r, \beta_r$ are sampled from the prior distributions described in (Geng et al., 2017) and $e_t \sim \mathcal{N}(0, 0.01^2)$ is a noise term that accounts for randomness in the tumour growth.

To incorporate heterogeneity among patient responses, due to, for instance, gender or genetic factors Bartsch et al. (2007), the prior means for $\beta_c$ and $\alpha_r$ are adjusted to create three patient subgroups $S^{(i)} \in \{1, 2, 3\}$ as described in Lim et al. (2018). This way, we incorporate in the model of tumour growth specific characteristics that affect the patient's individualized response to treatments. Thus, the prior mean $\mu_{\beta_c}$ of $\beta_c$ and the prior mean $\mu_{\alpha_r}$ of $\alpha_r$ are augmented as follows.

$$\mu'_{\beta_c}(i) = \begin{cases} 1.1\mu_{\beta_c}, & \text{if } S^{(i)} = 3 \\ \mu_{\beta_c}, & \text{otherwise} \end{cases} \qquad \mu'_{\alpha_r}(i) = \begin{cases} 1.1\mu_{\alpha_r}, & \text{if} S^{(i)} = 1 \\ \mu_{\alpha_r}, & \text{otherwise} \end{cases} \tag{21}$$

where $\mu_{\beta_c}$ and $\mu_{\alpha_r}$ are the mean parameters from Geng et al. (2017) and $\mu'_{\beta_c}(i)$ and $\mu'_{\alpha_r}(i)$ are the parameters used in the data simulation. The patient subgroup $S^{(i)} \in \{1, 2, 3\}$ is used as baseline features.

The chemotherapy drug concentration follows an exponential decay with half life of 1 day:

$$C(t) = \tilde{C}(t) + C(t-1)/2, \tag{22}$$

where $\tilde{C}(t) = 5.0mg/m^3$ of Vinblastine if chemotherapy is given at time $t$. $d(t) = 2.0Gy$ fractions of radiotherapy if the radiotherapy treatment is applied at timestep $t$.

Time-varying confounding is introduced by modelling chemotherapy and radiotherapy assignment as Bernoulli random variables, with probabilities $p_c$ and $p_r$ depending on the tumour diameter:

$$p_c(t) = \sigma\left(\frac{\gamma_c}{D_{\max}}(\bar{D}(t) - \delta_c)\right) \qquad p_r(t) = \sigma\left(\frac{\gamma_r}{D_{\max}}(\bar{D}(t) - \delta_r)\right), \tag{23}$$

where $\bar{D}(t)$ is the average tumour diameter over the last 15 days, $D_{\max} = 13$cm is the maximum tumour diameter and $\sigma(\cdot)$ is the sigmoid activation function. The parameters $\delta_c$ and $\delta_r$ are set to $\delta_c = \delta_r = D_{\max}/2$ such that there is 0.5 probability of receiving treatment when tumour is half of its maximum size. $\gamma_c, \gamma_r$ control the amount of time-dependent confounding; the higher $\gamma_\star$ is, the more important the history of tumour diameter is in assigning treatments. Thus, at each timestep, there are four treatment options options: no treatment ($A_1$), chemotherpy ($A_2$), radiotherapy ($A_3$), combined chemotherapy and radiotherapy ($A_4$).

Since the work most relevant to ours is the one of Lim et al. (2018) we used the same data simulation and same settings for $\gamma = \gamma_c = \gamma_r$ as in their case. When $\gamma = 0$, there is no time-dependent confounding and the treatments are randomly assigned. By increasing $\gamma$ we increase the influence of the volume size history (encoded in $\bar{D}(t)$) on the treatment probability. For example, assume $\bar{D}(t) = \frac{3D_{max}}{4}$. From equation (7), the probability of chemotherapy in this case is $p_c(t) = \sigma(\frac{\gamma_c}{D_{max}}(\bar{D}(t) - \frac{D_{max}}{2})) = \sigma(0.25\gamma_c)$, where $\sigma(\cdot)$ is the sigmoid function. When $\gamma = 1$, $p_c(t) = 0.56$, when $\gamma = 5$, $p_c(t) = 0.77$ and when $\gamma = 10$, $p_c(t) = 0.92$ in this example. $\gamma$ can be increased further to increase the bias. However, the values used in the experiments evaluate the model on a wide range of settings for the time-dependent confounding bias.

# G  MARGINAL STRUCTURAL MODELS

Marginal Structural Models (Robins et al., 2000; Hernán et al., 2001) have been widely used in epidemiology and as part of follow up studies. In our case, we would like to estimate the effects of a sequence of treatments in the future given the current patient history:

$$\mathbb{E}(\mathbf{Y}_{t+\tau} \mid \bar{\mathbf{A}}(t, t + \tau - 1) = \bar{\mathbf{a}}(t, t + \tau - 1), \bar{\mathbf{H}}_t) = g(\tau, a(t, t + \tau - 1), \bar{\mathbf{H}}_t), \qquad (24)$$

where $g$ is a generic function and $\bar{\mathbf{a}}(t, t + \tau - 1) = [\mathbf{a}_t, \ldots \mathbf{a}_{t+\tau-1}]$ represents a possible sequence of treatments from timestep $t$ just until before the potential outcome $\mathbf{Y}_{t+\tau}$ is observed. After removing the bias form time-dependent confounders, $\mathbb{E}(\mathbf{Y}_{t+\tau} \mid \bar{\mathbf{A}}(t, t + \tau - 1) = \bar{\mathbf{a}}(t, t + \tau - 1), \bar{\mathbf{H}}_t) = \mathbb{E}(\mathbf{Y}_{t+\tau}[\bar{\mathbf{a}}(t, t + \tau - 1)])$.

Note that for implementing MSMs, we encode the treatments at timestep $t$ in the model of tumour growth as $\mathbf{A}_t = [A_{t,c}, A_{t,d}]$ to indicate the binary application of chemotherapy and radiotherapy. In order to remove the time-dependent confounding bias and estimate future outcomes, we use the stabilized weights of MSMs to weight each patient in the dataset:

$$SW(t, \tau) = \prod_{n=t}^{t+\tau} \frac{f(\mathbf{A}_n \mid \bar{\mathbf{A}}_{n-1})}{f(\mathbf{A}_n \mid \bar{\mathbf{A}}_{n-1}, \bar{\mathbf{X}}_n, \mathbf{V})} = \prod_{n=t}^{t+\tau} \frac{\prod_{k \in \{c,d\}} f(A_{n,k} \mid \bar{\mathbf{A}}_{n-1})}{\prod_{k \in \{c,d\}} f(A_{n,k} \mid \bar{\mathbf{A}}_{n-1}, \bar{\mathbf{X}}_n, \mathbf{V})}, \qquad (25)$$

where $f(\cdot)$ represents the conditional probability mass function for discrete treatments.

We adopt the implementation in (Hernán et al., 2001; Howe et al., 2012; Lim et al., 2018) for MSMs and use logistic regression for estimating the propensity weights as follows:

$$f(A_{t,k} \mid \bar{\mathbf{A}}_{t-1}) = \sigma\Big(\sum_{j=1}^{k} \omega_k(\sum_{i=1}^{t-1} A_{t,j})\Big) \qquad (26)$$

$$f(A_{t,k} \mid \bar{\mathbf{H}}_t) = \sigma\Big(\sum_{k \in \{c,d\}} \phi_k(\sum_{i=1}^{t-1} A_{t,k}) + \mathbf{w}_1 \mathbf{X}_t + \mathbf{w}_2 \mathbf{X}_{t-1} + \mathbf{w}_3 \mathbf{V}\Big) \qquad (27)$$

where $\omega_\star, \phi_\star$ and $\mathbf{w}_\star$ are regression coefficients, $k \in \{c, d\}$ indicates the chemotherapy or radiotherapy treatments and $\sigma(\cdot)$ is the sigmoid function.

For predicting the outcome, the following regression model is used, where each individual patient is weighted by its propensity score:

$$g(\tau, a(t, t + \tau - 1), \bar{\mathbf{H}}_t) = \sum_{k \in \{c,d\}} \beta_k(\sum_{n=t}^{t+\tau-1} A_{n,k}) + \mathbf{l}_1 \mathbf{X}_t + \mathbf{l}_2 \mathbf{X}_{t-1} + \mathbf{l}_3 \mathbf{V} \qquad (28)$$

where $\beta_\star$ and $\mathbf{l}_\star$ are regression coefficients.

MSMs do not require hyperparameter tuning so we use the patients from both the train and validation sets for training.

# H  RECURRENT MARGINAL STRUCTURAL NETWORKS

MSMs are very sensitive to model mis-specification in computing the propensity weights and estimating the outcomes. Recurrent Marginal Structural Models (RMSNs) (Lim et al., 2018) overcome this problem by using recurrent neural networks to estimate the propensity scores and to build the outcome model. RNNs are more robust to changes in the treatment assignment policy. RMSNs were implemented as descried in Lim et al. (2018)[2].

For implementing RMSNs, we also encode the treatments at timestep $t$ in the model of tumour growth as $\mathbf{A}_t = [A_{t,c}, A_{t,d}]$ to indicate the binary application of chemotherapy and radiotherapy. The propensity weights are estimated using recurrent neural networks as follows:

$$f(A_{t,k} \mid \bar{\mathbf{A}}_{t-1}) = \text{RNN}_{SW_n}(\bar{\mathbf{A}}_{t-1}) \qquad f(A_{t,k} \mid \bar{\mathbf{X}}_t, \bar{\mathbf{A}}_{t-1}) = \text{RNN}_{SW_d}(\bar{\mathbf{A}}_{t-1}, \bar{\mathbf{X}}_t, \mathbf{V}) \qquad (29)$$

---

[2]We used the publicly available implementation from `https://github.com/sjblim/rmsn_nips_2018`.

For predicting one-step-ahed outcome, R-MSNs use an encoder network:

$$g(1, a(t,t), \bar{\mathbf{H}}_t) = \text{RNN}_E(\mathbf{a}_t, \bar{\mathbf{A}}_{t-1}, \bar{\mathbf{X}}_t, \mathbf{V}), \tag{30}$$

where in the loss function, each patient is weighted by their stabilized IPTW.

For estimating the treatment responses for a sequence of treatments in the future, RMSNs use a decoder network:

$$g(\tau, a(t, t + \tau - 1), \bar{\mathbf{H}}_t) = \text{RNN}_D(\mathbf{a}_t, \ldots, \mathbf{a}_{t+\tau-1}, \bar{\mathbf{A}}_{t-1}, \bar{\mathbf{X}}_t, \mathbf{V}). \tag{31}$$

See Lim et al. (2018) for more details about the R-MSNs model architecture and training procedure of the propensity weights, encoder and decoder networks. Tables 2 and 3 show the hyperparameter search ranges used to optimize this model for evaluation in our paper. The hyperparameters were selected in the same way as proposed by Lim et al. (2018), based on the error on the factual outcomes in the validation dataset. All of the models are trained using Adam optimizer for 100 epochs.

Table 2: Hyperparameter search range for propensity networks and encoder (same as in Lim et al. (2018)). C is the size of the input.

| Hyperparameter | Search range |
| --- | --- |
| Iterations of Hyperparameter Search | 50 |
| Learning rate | 0.01, 0.005, 0.001 |
| Minibatch size | 64, 128, 256 |
| RNN state size | 0.5C, 1C, 2C, 3C, 4C |
| Dropout rate | 0.1, 0.2, 0.3, 0.4, 0.5 |
| Max Gradient Norm | 0.5, 1.0, 2.0 |

Table 3: Hyperparameter search range for decoder (same as in Lim et al. (2018)). C is the input size.

| Hyperparameter | Search range |
| --- | --- |
| Iterations of Hyperparameter Search | 20 |
| Learning rate | 0.01, 0.001, 0.0001 |
| Minibatch size | 256, 512, 1024 |
| RNN state size | 1C, 2C, 4C, 8C, 16C |
| Dropout Rate | 0.1, 0.2, 0.3, 0.4, 0.5 |
| Max Gradient Norm | 0.5, 1.0, 2.0, 4.0 |

## I   BASELINE RNN AND LINEAR MODEL

For the baseline linear model, we fit the same regression model used for Marginal Structural Networks, but without using the IPTW. The baseline RNN uses an LSTM unit and, at each timestep, receives as input the current treatment, the patient covariates and the patient static features to perform one-step-ahead prediction. To have a model of similar capacity to the CRN (similar number of parameters), we add a fully connected layer on top of the output of the LSTM unit in order to obtain the outcomes. Table 4 shows the hyperparameter search range used to optimize this model. The hyperparameters were selecting according to the error on the factual outcomes in the validation set. We train the baseline RNN using the Adam optimizer for 100 epochs.

Table 4: Hyperparameter search range for baseline RNN model. C is the size of the input.

| Hyperparameter | Search range |
|---|---|
| Iterations of Hyperparameter Search | 50 |
| Learning rate | 0.01, 0.001, 0.0001 |
| Minibatch size | 64, 128, 256 |
| RNN hidden units | 0.5C, 1C, 2C, 3C, 4C |
| FC hidden units | 0.5C, 1C, 2C, 3C, 4C |
| RNN dropout probability | 0.1, 0.2, 0.3, 0.4, 0.5 |

## J  HYPERPARAMETER OPTIMIZATION FOR CRN

As described in Appendix C, the dataset for training the decoder are used by splitting the sequences of the patients in the training set. This creates a larger dataset for training (where each patient $(i)$ contributes $T^{(i)} - \tau_{\max}$ times to the dataset) which requires a different hyperparameter search range. Moreover, the balancing representations computed by the encoder are used to initialize the state of the RNN for the decoder. Thus, the decoder RNN size is equal to the size of the balancing representation size of the encoder. Table 5 shows the hyperparameter search ranges for the encoder and decoder networks in CRN. We selected hyperparameters based on the error of the model on the factual outcomes in the validation dataset. All models are trained for 100 epochs.

In addition, Tables 6 and 7 illustrate the optimal hyperparameters chosen.

Table 5: Hyperparameter search range for CRN encoder. C is the size of the input and R is the size of the balancing representation.

| Hyperparameter | Search range encoder | Search range decoder |
|---|---|---|
| Iterations of Hyperparameter Search | 50 | 30 |
| Learning rate | 0.01, 0.001, 0.0001 | 0.01, 0.001, 0.0001 |
| Minibatch size | 64, 128, 256 | 256, 512, 1024 |
| RNN hidden units | 0.5C, 1C, 2C, 3C, 4C | Balancing representation size of encoder |
| Balancing representation size | 0.5C, 1C, 2C, 3C, 4C | 0.5C, 1C, 2C, 3C, 4C |
| FC hidden units | 0.5R, 1R, 2R, 3R, 4R | 0.5R, 1R, 2R, 3R, 4R |
| RNN dropout probability | 0.1, 0.2, 0.3, 0.4, 0.5 | 0.1, 0.2, 0.3, 0.4, 0.5 |

Table 6: Optimal hyperparameters for the CRN encoder when different degrees of time-dependent confounding are applied in the model of tumour growth. The parameters $\gamma_c$ and $\gamma_r$ measure the degree of time-dependent confounding applied. When $\gamma_c$ and $\gamma_r$ are set to the same value, we denote this with $\gamma_\star$.

|  | $\gamma_\star = 0$ | $\gamma_\star = 1$ | $\gamma_\star = 2$ | $\gamma_\star = 3$ | $\gamma_\star = 4$ | $\gamma_\star = 5$ |
|---|---|---|---|---|---|---|
| Learning rate | 0.001 | 0.1 | 0.001 | 0.01 | 0.01 | 0.001 |
| Minibatch size | 64 | 64 | 64 | 128 | 64 | 128 |
| RNN hidden units | 12 | 18 | 24 | 18 | 24 | 24 |
| Balancing representation size | 18 | 18 | 12 | 18 | 6 | 12 |
| FC hidden units | 18 | 18 | 36 | 54 | 24 | 48 |
| RNN dropout probability | 0.1 | 0.1 | 0.1 | 0.2 | 0.2 | 0.1 |
|  | $\gamma_\star = 6$ | $\gamma_\star = 7$ | $\gamma_\star = 8$ | $\gamma_\star = 9$ | $\gamma_\star = 10$ |  |
| Learning rate | 0.001 | 0.001 | 0.01 | 0.001 | 0.01 |  |
| Minibatch size | 64 | 64 | 128 | 128 | 128 |  |
| RNN hidden units | 24 | 18 | 12 | 24 | 24 |  |
| Balancing representation size | 12 | 18 | 24 | 18 | 12 |  |
| FC hidden units | 48 | 72 | 12 | 36 | 12 |  |
| RNN dropout probability | 0.1 | 0.2 | 0.1 | 0.1 | 0.1 |  |
|  | $\gamma_c = 0, \gamma_r = 5$ | $\gamma_c = 5, \gamma_r = 0$ |  |  |  |  |
| Learning rate | 0.01 | 0.001 |  |  |  |  |
| Minibatch size | 128 | 64 |  |  |  |  |
| RNN hidden units | 12 | 12 |  |  |  |  |
| Balancing representation size | 18 | 24 |  |  |  |  |
| FC hidden units | 36 | 96 |  |  |  |  |
| RNN dropout probability | 0.1 | 0.1 |  |  |  |  |

Table 7: Optimal hyperparameters for the CRN decoder when different degrees of time-dependent confounding are applied in the model of tumour growth. The parameters $\gamma_c$ and $\gamma_r$ measure the degree of time-dependent confounding applied. When $\gamma_c$ and $\gamma_r$ are set to the same value, we denote this with $\gamma_\star$

|  | $\gamma_\star = 1$ | $\gamma_\star = 2$ | $\gamma_\star = 3$ | $\gamma_\star = 4$ | $\gamma_\star = 5$ |
|---|---|---|---|---|---|
| Learning rate | 0.001 | 0.001 | 0.001 | 0.001 | 0.001 |
| Minibatch size | 1024 | 1024 | 512 | 1024 | 1024 |
| RNN hidden units | 18 | 12 | 18 | 6 | 12 |
| Balancing representation size | 18 | 18 | 6 | 18 | 3 |
| FC hidden units | 18 | 36 | 18 | 72 | 6 |
| RNN dropout probability | 0.1 | 0.2 | 0.3 | 0.1 | 0.1 |
|  | $\gamma_c = 0, \gamma_r = 5$ | $\gamma_c = 5, \gamma_r = 0$ |  |  |  |
| Learning rate | 0.01 | 0.001 |  |  |  |
| Minibatch size | 512 | 1024 |  |  |  |
| RNN hidden units | 18 | 24 |  |  |  |
| Balancing representation size | 18 | 12 |  |  |  |
| FC hidden units | 36 | 24 |  |  |  |
| RNN dropout probability | 0.1 | 0.03 |  |  |  |

# K  FULL RESULTS FOR COUNTERFACTUAL PREDICTION

## K.1  MULTI-STEP AHEAD PREDICTION OF COUNTERFACTUALS

Figure 7 shows the normalized RMSE for multiple step-ahead prediction of counterfactuals. The RMSE is normalized by the maximum tumour volume: $V_{max} = 1150cm^3$. The counterfactuals in this case are generated as described in Section 6.3 and Appendix I. We notice that performance gains of CRN compared to RMSN increase with the number of future timesteps for which the counterfactuals are estimated.

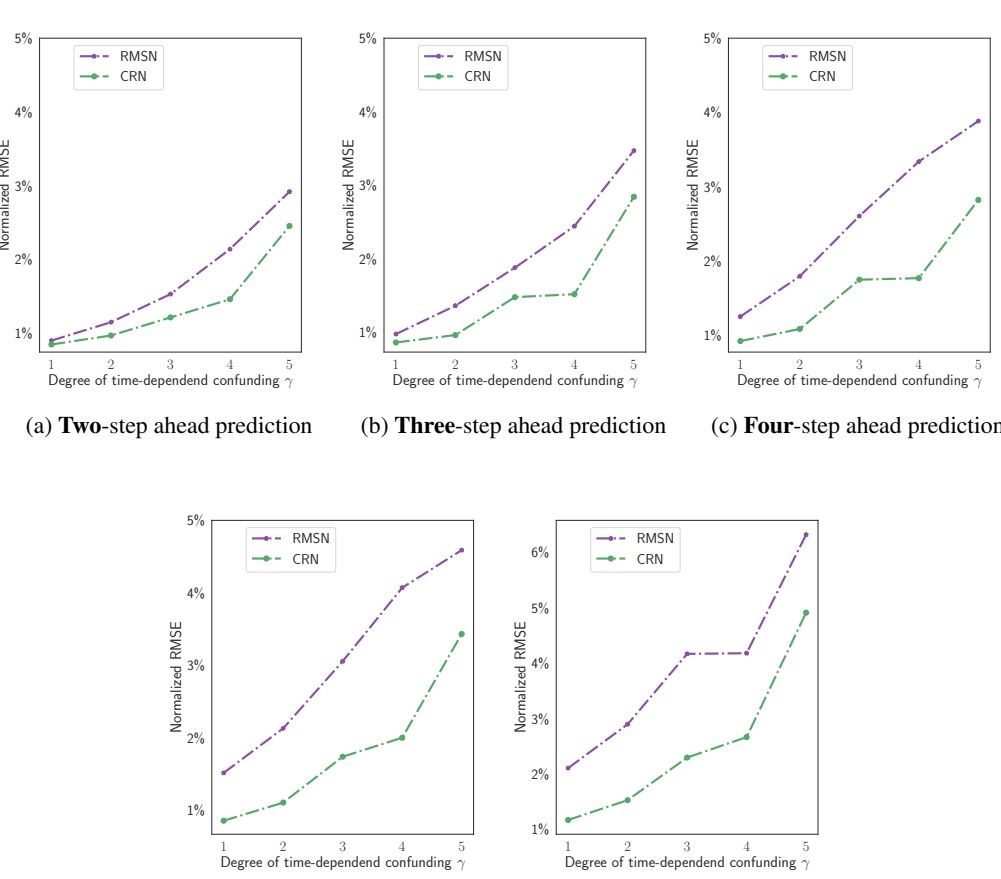

Figure 7: Results for prediction of patient counterfactuals for multiple steps ahead.

## K.2    DETAILED RESULTS FOR THE COUNTERFACTUAL PREDICTIONS

Tables 8 and 9 show detailed results for the counterfactual predictions.

Table 8: Normalized RMSE for one-step-ahead prediction of counterfactuals. The parameter $\gamma$ measures the degree of time-dependent confounding applied.

|  | $\gamma = 0$ | $\gamma = 1$ | $\gamma = 2$ | $\gamma = 3$ | $\gamma = 4$ | $\gamma = 5$ |
|---|---|---|---|---|---|---|
| Linear (no IPTW) | 0.99% | 1.08% | 1.36% | 1.68% | 2.11% | 2.77% |
| MSM | 0.99% | 1.08% | 1.34% | 1.63% | 2.02% | 2.61% |
| RNN | 0.70% | 0.70% | 0.84% | 1.05% | 1.24% | 1.69% |
| CRN ($\lambda = 0$) | 0.66% | 0.77% | 0.92% | 0.95% | 1.24% | 1.54% |
| RMSN | 0.60% | 0.61% | 0.72% | 0.81% | 0.94% | 1.23% |
| CRN | **0.56%** | **0.57%** | **0.62%** | **0.67%** | **0.87%** | **1.20%** |

|  | $\gamma = 6$ | $\gamma = 7$ | $\gamma = 8$ | $\gamma = 9$ | $\gamma = 10$ |  |
|---|---|---|---|---|---|---|
| Linear (no IPTW) | 3.55% | 4.15% | 4.80% | 5.09% | 5.22% |  |
| MSM | 3.30% | 3.79% | 4.30% | 4.47% | 4.47% |  |
| RNN | 2.03% | 2.52% | 2.88% | 3.79% | 4.01% |  |
| CRN ($\lambda = 0$) | 1.98% | 2.42% | 2.73% | 3.17% | 3.57% |  |
| RMSN | 1.70% | 2.18% | 2.37% | 2.77% | 2.83% |  |
| CRN | **1.48%** | **1.56%** | **2.05%** | **2.36%** | **2.41%** |  |

Table 9: Normalized RMSE for $\tau$-step-ahead prediction of counterfactuals. The parameter $\gamma$ measures the degree of time-dependent confounding applied.

|  |  | $\gamma = 1$ | $\gamma = 2$ | $\gamma = 3$ | $\gamma = 4$ | $\gamma = 5$ |
|---|---|---|---|---|---|---|
| $\tau = 2$ | RMSN | 0.90% | 1.15% | 1.53% | 2.14% | 2.91% |
|  | CRN | **0.84%** | **0.96%** | **1.21%** | **1.46%** | **2.45%** |
| $\tau = 3$ | RMSN | 0.97% | 1.36% | 1.87% | 2.44% | 3.47% |
|  | CRN | **0.86%** | **0.96%** | **1.47%** | **1.51%** | **2.84%** |
| $\tau = 4$ | RMSN | 1.24% | 1.79% | 2.60% | 3.33% | 3.88% |
|  | CRN | **0.91%** | **1.08%** | **1.74%** | **1.76%** | **2.82%** |
| $\tau = 5$ | RMSN | 1.51% | 2.13% | 3.06% | 4.07% | 4.58% |
|  | CRN | **0.85%** | **1.10%** | **1.73%** | **2.00%** | **3.43%** |
| $\tau = 6$ | RMSN | 2.10% | 2.89% | 3.06% | 4.16% | 6.32% |
|  | CRN | **1.16%** | **1.52%** | **2.29%** | **2.66%** | **4.91%** |

## L   TEST SET GENERATION FOR EVALUATING TIMING OF TREATMENT

In order to evaluate how well the models select the correct treatment and timing of treatment we simulate counterfactual outcomes as follows. We generate 1000 test samples using the model of tumour growth described in Section 6. Let $\bar{\mathbf{H}}_t$ be the current history of the patient and let $\tau$ be a future time horizon. For each timestep in the future, we have 4 treatment options at: no treatment ($A_0$), chemotherapy ($A_1$), radiotherapy ($A_2$), chemotherapy and radiotherapy. ($A_3$).

Using the model of tumour growth where the outcome $\mathbf{Y}_{t+\tau}$ is given by the volume of the tumour, we generate the following $2\tau$ counterfactuals:

Chemotherapy application

$$\mathbf{Y}_{t+\tau} \quad | \quad \mathbf{a}_t = A_1, \mathbf{a}_{t+1} = A_0, \ldots \mathbf{a}_{t+\tau-1} = A_0, \bar{\mathbf{H}}_t \tag{32}$$

$$\mathbf{Y}_{t+\tau} \quad | \quad \mathbf{a}_t = A_0, \mathbf{a}_{t+1} = A_1, \ldots \mathbf{a}_{t+\tau-1} = A_0, \bar{\mathbf{H}}_t \tag{33}$$

$$\ldots$$

$$\mathbf{Y}_{t+\tau} \quad | \quad \mathbf{a}_t = A_0, \mathbf{a}_{t+1} = A_0, \ldots \mathbf{a}_{t+\tau-1} = A_1, \bar{\mathbf{H}}_t \tag{34}$$

Radiotherapy application

$$\mathbf{Y}_{t+\tau} \quad | \quad \mathbf{a}_t = A_2, \mathbf{a}_{t+1} = A_0, \ldots \mathbf{a}_{t+\tau-1} = A_0, \bar{\mathbf{H}}_t \tag{35}$$

$$\mathbf{Y}_{t+\tau} \quad | \quad \mathbf{a}_t = A_0, \mathbf{a}_{t+1} = A_2, \ldots \mathbf{a}_{t+\tau-1} = A_0, \bar{\mathbf{H}}_t \tag{36}$$

$$\ldots$$

$$\mathbf{Y}_{t+\tau} \quad | \quad \mathbf{a}_t = A_0, \mathbf{a}_{t+1} = A_0, \ldots \mathbf{a}_{t+\tau-1} = A_2, \bar{\mathbf{H}}_t \tag{37}$$

We perform this for each patient in the test set and at each time $t$ in the history. For instance, for a patient with 50 timesteps in the model of tumour growth and for time horizon $\tau = 3$, we generate $2 \cdot 3 \cdot 50 = 300$ counterfactuals.

Using the true generated couterfactual data, we select the treatment that has the lowest $\mathbf{Y}_{t+\tau}$ among the $\tau$ options generated for each treatment. Then, we select the time of applying treatment (among $t, t+1, \ldots t+\tau-1$) that resulted in the lowest $\mathbf{Y}_{t+\tau}$. For each model, we generate the counterfactuals under the same treatment plans and patient histories. Then, we perform the selection of treatment and timing of treatment in the same way and we compare these with the true ones. Note that in order to account for numerical instability (two outcomes $Y_{t+\tau}$ having very similar values), we consider two outcomes the same if they are within $\epsilon = 0.001$ of each other.

## M    RESULTS ON FACTUAL PREDICTION ON MIMIC III

Using the Medical Information Mart for Intensive Care (MIMIC III) (Johnson et al., 2016) database consisting of electronic health records from patients in the ICU, we also show how the CRN can be used on a real medical dataset. From MIMIC III we extracted the patients on antibiotics, with trajectories up to 30 timesteps, thus obtaining a dataset with $3487$ patients. For each patient, we extracted $25$ patient covariates including lab tests and vital signs measured over time, as well as static patient features such as age and gender.

We used a binary treatment at each timestep indicating whether the patient was administered antibiotics or not. Note that for the longitudinal covariates we used aggregate value for each day since the ICU admission. The reason for this is because antibiotic treatment is decided daily for the patient. We split the dataset into $2826/313/348$ patients for training, validation and testing respectively. We performed hyperparameter optimization on the validation patient set, using the search ranges in Table 5 and we again selected hyperparameters based on the error on the factual outcomes.

We estimate the individualized effect of antibiotics assigned over time on the patient's white blood cell count. A high white blood cell count is associated with severe illness and poor outcome for ICU patients (Waheed et al., 2003). Antibiotic administration in the ICU aims to reduce the white blood cell count. However, the effectiveness of the antibiotics treatment in reducing the white blood cell count is highly dependent on the time they are administered with respect to the history of the patient covariates. In this context we again have time-dependent confounders: the patient features change over time and are affected by the previous administration of antibiotics. Moreover, the history of the patient features also determines antibiotics administration and affects future patient outcomes (De Bus et al., 2018; Ali et al., 2019).

In Table 10 we report the root mean squared error for factual prediction of the patients' white blood cell count for multiple prediction horizons $\tau$. Note that for this dataset we do not have access to counterfactual data, which is why we report error on factual predictions.

Table 10: RMSE for $\tau$-step-ahead prediction of factual outcomes on MIMIC III.

|      | $\tau = 1$ | $\tau = 2$ | $\tau = 3$ | $\tau = 4$ |
|------|------------|------------|------------|------------|
| RMSN | 2.84       | 3.87       | 4.46       | 4.79       |
| CRN  | 2.68       | 3.54       | 4.07       | 4.67       |

We notice that CRN also achieves better performance than RMSN in estimating factual outcomes in a real-world dataset containing electronic health records. In this context, where couterfactual data is not available, domain expert knowledge is required to validate the model's counterfactual predictions under other antibiotic treatment alternatives. This further medical validation is outside the scope of this paper.

