# OpenReview forum: "Estimating counterfactual treatment outcomes over time through adversarially balanced representations"
_ICLR.cc/2020/Conference — Accept (Spotlight)_

### Official Review · AnonReviewer2 · 2019-10-15
**Official Blind Review #2**

**Rating:** 6

**Review:**

The paper adapts domain adversarial training to construct treatment invariant representation for adjusting time-varying confounding in counterfactual treatment outcome predictions over time.

The proposed method extends the recent static settings of balancing representation in counterfactual inference to longitudinal settings, and it also overcomes the problem of high-variance of IPTW weighting in MSN based methods such as the current state-of-art method RMSN.

The paper is very well written. Method is novel to me. Experiments are sufficient. However since I am not very familiar with this area, so there can be things I miss in the evaluation of this paper.

There is one problem with the illustrations in Figure 1: predictions on the potential outcomes before the first treatment was given should have stayed on the same path, and then depart when different treatment was initiated (at different time)

**Experience Assessment:**

I have read many papers in this area.

**Review Assessment: Checking Correctness Of Derivations And Theory:**

I did not assess the derivations or theory.

**Review Assessment: Checking Correctness Of Experiments:**

I did not assess the experiments.

**Review Assessment: Thoroughness In Paper Reading:**

I read the paper at least twice and used my best judgement in assessing the paper.

---

> ### Author Response · Authors · 2019-11-07
> **Reply for Reviewer 2**
>
> Thank you very much for your review!
>
> Thank you for the comment regarding the illustration in Figure 1! We thought that the current illustration would make it more clear on which counterfactual path the treatment was given. However, we do agree with your remark and we will incorporate it to make the figure more accurate!

---

### Official Review · AnonReviewer3 · 2019-10-22
**Official Blind Review #3**

**Rating:** 6

**Review:**

The paper introduces Counterfactual Recurrent Network (CRN) that is able to estimate the effects of various treatments from longitudinal data. The claim is that the model can decide (i) treatment plan; (ii) optimal time of treatment; and (iii) when to stop treatment. The proposed method attempts to learn time-invariant representations that are not predictive of the next treatment by borrowing ideas from Ganin, et al. (2016)’s work on for domain adversarial training. In fact, this paper is an extension of (Atan et al., 2018) to be applicable for longitudinal data.

This paper should be rejected due to the following arguments:
	- Page 1, par. 4, line 4-7: The example given in sentence “For instance, if ...” is true regardless of time. Therefore, it does not describe a situation where a time-variant confounder might make causal inference hard. In fact, nowhere in the paper there is a solid description of how time-variant confounders would break the current approaches for causal inference.

	- Page 3, par. 4, last sentence: I don’t understand why sequential treatments and change of covariates through time can stop us from using the conventional methods for learning balanced representations. Why not consider each time point as an instance (thus representing each patient with multiple instances) and learn the representation from the collection of all these instances?
All in all, I don’t understand why the authors assume that the world is non-Markovian and that they have to consider the entire \bar{H}_t. Especially when in the their Experiments section the model used for tumor growth that generates the synthetic data is Markov, and consequently, does not check if the proposed method works in a non-Markovian world.
In summary, if the goal is to solve the non-Markov case, the presented experiments don’t show that the proposed method is addressing that; and if the  goal is to solve the Markov case: (i) why insist on framing the problem as non-Markov; and (ii) (Atan et al., 2018) already have solved this (since we established that there are no time-dependent confounders and as the authors state on page 4, par. 3, line -4 to -3 “the novelty here comes from the use of domain adversarial training to eliminate bias from the time-dependent confounders”).

	- Page 6, Theorem 1 and 2 lines above it: This theorem does not prove that the confounding bias is completely removed because the objective function in Eq. (4) being optimized also includes a loss term for outcome prediction. In fact, IF there is a confounding bias (that [partially] determines both treatment and outcome), it would be wrong to remove it from the learned representation.

Things to improve the paper that did not impact the score:
	- Page 2, par. 1, line -2: Use of adverb “Moreover” is wrong. Extreme weights are due to division by small Pr(t|x). The consequence of this numerical instability is high-variance estimates.
	- Page 2, par. 4, 1st sentence: incomprehensible
	- Page 2, par. 4, line -2: “but also” should come after a “not only”.
	- Page 3, par. 4, line -1: incorrect usage of citation; both author name(s) and publication year must be in parentheses.
	- Page 4, par. 5, last line: sentence “For testing, the … “ needs elaboration.
	- Page 6, par. 1, line -3: “start of” should be “start off”.

References:
	- Atan, O., Zame, W. R., & Van Der Schaar, M. (2018). Counterfactual Policy Optimization Using Domain-Adversarial Neural Networks. In ICML CausalML workshop.
	- Ganin, Y., Ustinova, E., Ajakan, H., Germain, P., Larochelle, H., Laviolette, F., ... & Lempitsky, V. (2016). Domain-adversarial training of neural networks. The Journal of Machine Learning Research, 17(1), 2096-2030.


********UPDATE after reading the rebuttal********
The authors have addressed my major concerns in their rebuttal and therefore I have increased my rating form “reject” to “weak accept”.



**Experience Assessment:**

I have published one or two papers in this area.

**Review Assessment: Checking Correctness Of Derivations And Theory:**

I carefully checked the derivations and theory.

**Review Assessment: Checking Correctness Of Experiments:**

I carefully checked the experiments.

**Review Assessment: Thoroughness In Paper Reading:**

I read the paper thoroughly.

---

> ### Author Response · Authors · 2019-11-07
> **Initial reply for Reviewer 3**
>
> Thank you very much for your review!  We would like to clarify your concerns about our work.
>
>
> * Related Work *
>
> We highlight below the differences between our work and the work of (Atan et al., 2018) (these were also described in Appendix A). Please note that (Atan et al., 2018) addresses the problem of policy optimization in the static setting which is different from causal inference.
>
> 1. While (Atan et al., 2018) also use domain adversarial training to remove confounding bias in the static setting to infer optimal policies, their domains have a different meaning: the entire observational data represents the source domain and a simulated dataset where the actions (treatments) are assigned randomly represents the target domain. Through domain adversarial training they build a representation that is invariant with respect to these two domains (See Section 5.1 in (Atan et al., 2018)). In our work, the different treated populations at each time-step represent the different domains and our aim is to build a representation that is invariant with respect to the treatment received by the patient at each timestep.
>
> 2. Directly extending the method proposed by (Atan et al., 2018) to handle time-varying confounders and actions would require generating random sequences of actions over time (to create their target domain), which is impractical.
>
> 3. Our model performs causal inference and can be used to estimate counterfactual outcomes under intended treatments plans. Once these counterfactual outcomes are estimated, they can be used to decide optimal treatment plans and timings of treatments. Policy optimization methods, like the one in (Atan et al., 2018) directly predict an optimal treatment from the data and cannot be used for causal inference.
>
>
> * Time-dependent confounders *
>
> Time-dependent confounders are confounders that change over time depending on past treatments and that also influence future treatment assignments and outcomes. Time-dependent confounders are certainly present in observational data because treatments are generally adjusted based on the patients’ response [1].
>
> This requires the world to be modelled as non-Markovian and for the patient history to be taken into account to be able to model how the changes in patient covariates over time affect the patient outcomes under different treatment plans.
>
> Current approaches for causal inference in the static setting are aimed at handling the cross-sectional set-up, where the treatment and outcome depend only on a static value of the patient covariates. If we consider each time point as an instance and represent the patient by multiple such instances, we cannot model how that patient’s history which contains the time-dependent confounders affects the patient outcome. The methods that propose balancing representations for the static causal inference setting only take into account static patient features and not patient histories.
>
> Please also refer to Appendix C where we illustrate the causal graph for time-dependent confounders and how they affect treatment assignments and outcomes. In Appendix C, we also provide an explanation based on the causal graphs about how the CRN reduces the time-dependent confounding bias.
>
>
> * Theorem 1 *
>
> Theorem 1 shows that the treatment loss part of the objective aims to remove the confounding bias. The outcome loss part of the objective ensures that the representation preserves the information predictive of the patient outcome. The few lines after Theorem 1 in our paper also emphasize that the aim of our training objective is to find a representation that minimizes the imbalance between the treatment distributions and that can be used for estimating treatment outcome.
>
> A similar approach for removing confounding bias through balanced representations was used by Johanson et al. [2] in the static causal inference setting. Their proposed method for building the representation uses discrepancy measures (such as the Maximum Mean Discrepancy) in the loss function to make treated and control distributions similar. This does not directly extend to multiple treatments (as also noted by the authors) and does not model patient history and time-dependent confounders. In our work, we measure the discrepancy between distributions based on their separability by a discriminatively-trained classifier which is directly applicable to multiple treatments. In addition, our model architecture handles the time-dependent confounders.
>
>
> ***
>
> We will also address the comments that you have mentioned for improving the paper.
>
> We hope that our reply has addressed your concerns. We are very happy to provide further clarifications if needed!
>
> [1] Mohammad Ali Mansournia, Mahyar Etminan, Goodarz Danaei, Jay S Kaufman, and Gary Collins. Handling time-varying confounding in observational research. BMJ, 2017
>
> [2] Fredrik Johansson, Uri Shalit, and David Sontag. Learning representations for counterfactual inference. ICML, 2016

---

> > ### Author Response · Authors · 2019-11-07
> > **Additional reply for Reviewer 3 about experiments**
> >
> > We would also like to clarify your comments about the experiments and provide you with more results. We would be happy to perform more experiments that you consider necessary.
> >
> >
> > * Experiments *
> >
> > Regarding our experimental set-up, please note that the treatment assignments (bottom of page 6 and equation (23) in Appendix F) depend on the mean value of the history of the tumour diameter for the past 15 days. This introduces time-dependent confounding since the history of the tumour diameter (and not just the current value) affects the treatment assignment and subsequently the future outcomes.
> >
> > In addition, the chemotherapy drug concentration C(t) follows an exponential decay and depends on the current chemotherapy treatment assignment and past drug concentration (equation (22)). We provide in Appendix F a full description of the tumour growth simulation.
> >
> > Please also note that the same data simulation was used to evaluate Recurrent Marginal Structural Networks (Lim et al., NeurIPS 2018) [3] a model that uses the inverse probability of treatment weighting to handle the time-dependent confounders and estimate patient outcomes.
> >
> >
> > * Additional experimental results *
> >
> > To further improve the experiments and illustrate the applicability of the Counterfactual Recurrent Network in more complex scenarios involving real data, we also performed experiments using the Medical Information Mart for Intensive Care (MIMIC III) database consisting of electronic health records from patients in the ICU.
> >
> > From MIMIC III we extracted a dataset of 3487 patients on antibiotics (treatments). For each patient, we extracted 25 patient covariates including laboratory tests and vital signs measured over time, as well as static patient features such as age and gender. We modelled the antibiotics as binary treatments indicating whether the patient has received antibiotics or not at each timestep.
> >
> > We train the Counterfactual Recurrent Network (CRN) and Recurrent Marginal Structural Networks (RMSNs) models to estimate the effect of the antibiotics treatment on the white blood cell count. In this context we again have time-dependent confounders: the patient features change over time and are affected by the previous administration of antibiotics. Moreover, the history of the patient features also determines antibiotics administration and affects future patient outcomes. [4, 5]
> >
> > In the following table, we report the root mean squared error for factual prediction of the patients' white blood cell count for multiple prediction horizons tau. Note that for this dataset we do not have access to counterfactual data, which is why we only report the error on factual outcomes (corresponding to the treatments observed in the dataset).
> >
> > -----------------------------------------------------------------------------------------------
> >  Model                   |      tau = 1     |     tau = 2     |     tau = 3    |   tau = 4
> > ------------------------------------------------------------------------------------------------
> >  RMSN                   |        2.84        |       3.87        |     4.46         |     4.79
> >  CRN                      |        2.68        |       3.54        |     4.07         |     4.67
> > -----------------------------------------------------------------------------------------------
> >
> > We will update the paper to include a detailed description of the data used and of these experimental results.
> >
> > [3] Bryan Lim, Ahmed Alaa, and Mihaela van der Schaar. Forecasting treatment responses over time using recurrent marginal structural networks. NeurIPS, 2018.
> >
> > [4] De Bus, Liesbet, et al. "A complete and multifaceted overview of antibiotic use and infection diagnosis in the intensive care unit: results from a prospective four-year registration." Critical Care 22.1 (2018)
> >
> > [5] Ali, Muhammad, et al. "Rational use of antibiotics in an intensive care unit: a retrospective study of the impact on clinical outcomes and mortality rate." Infection and Drug Resistance 12 (2019): 493.

---

### Official Review · AnonReviewer1 · 2019-10-23
**Official Blind Review #1**

**Rating:** 8

**Review:**

This work addresses the problem of causal inference in time-dependent treatment regimes. To address the problem, the authors propose an extension of the balancing representation for causal inference framework that seeks render the current treatment independent from a representation of the history of treatment and confounders. This is sensibly actualized within an RNN. The authors provide empirical results that demonstrate the proposed method performing very well in comparison to prior art.

This paper is well written, proposes a sensible solution to a difficult problem, and performs well empirically. It would be nice to see theory that connects the imbalance to the expected error of the causal estimand, and a notion of hyperparameter tuning (the authors say they search over hyperparameters but do not specify how the parameters are ultimately selected. However, I don't feel that either of those points outweigh the benefit of the paper.

**Experience Assessment:**

I have published one or two papers in this area.

**Review Assessment: Checking Correctness Of Derivations And Theory:**

I carefully checked the derivations and theory.

**Review Assessment: Checking Correctness Of Experiments:**

I assessed the sensibility of the experiments.

**Review Assessment: Thoroughness In Paper Reading:**

I read the paper thoroughly.

---

> ### Author Response · Authors · 2019-11-07
> **Reply for Reviewer 1**
>
> Thank you very much for your review!
>
> We have selected hyperparameters according to the model error on the factual outcomes in the validation dataset. The hyperparameters were chosen in the same way by Lim et al. for Recurrent Marginal Structural Networks [1]. We will make this more clear in the paper.
>
> We agree with you that this might not be the ideal way of selecting hyperparameters. However, hyperparameter selection is an open problem in causal inference where the counterfactual outcomes are never observed. Moreover, some other approaches used for hyperparameter selection in the static setting, such as nearest neighbour matching [2] are not directly applicable to the longitudinal setting due to the difficulty of matching entire patient trajectories to find approximations for the counterfactuals.
>
> Despite its wide applicability, the estimation of treatment responses in a time-series setting has been a lot less studied compared to the static case of binary treatments. In future work, we will aim to improve the balancing representations and work on providing theoretical guarantees for the error on the counterfactuals. For completeness, will include a discussion in the paper of these future directions for this work.
>
> [1] Bryan Lim, Ahmed Alaa, and Mihaela van der Schaar. Forecasting treatment responses over time using recurrent marginal structural networks. NeurIPS, 2018.
> [2] Uri Shalit, Fredrik D Johansson, and David Sontag. Estimating individual treatment effect: Generalization bounds and algorithms. ICML, 2017.

---

### Author Response · Authors · 2019-11-10
**Revised paper**

We have updated the paper to incorporate the constructive feedback provided by the reviewers.

In Appendix M we have added the additional results on the MIMIC III database consisting of electronic health records for patients in the ICU. These results show the applicability of our model in a more complex, real-world scenario.

Reviewer 1: We have updated the paper to explicitly mention how hyperparameters are selected for each benchmark model (see Appendices H, I, J). In addition, in Section 7 we have incorporated a discussion of future work (see the text added in green).

Reviewer 2: We have updated Figure 1 according to your feedback.

Reviewer 3: We have improved the example in the introduction to explain better how time-dependent confounders break current approaches for causal inference in the static setting. Please see the text added in blue in the introduction. We have also addressed the minor comments that you provided for improving the paper.

Please let us know if further clarifications are needed.

---

### Decision · Program_Chairs · 2019-12-19

**Decision:**

Accept (Spotlight)

**Comment:**

Reviewers uniformly suggest acceptance. Please look carefully at reviewer comments and address in the camera-ready. Great work!